# Real-world performance of SARS-Cov-2 serology tests in the United States, 2020

**Carla V. Rodriguez-Watson**[1]*, **Anthony M. Louder**[2], **Carly Kabelac**[2], **Christopher M. Frederick**[3], **Natalie E. Sheils**[4], **Elizabeth H. Eldridge**[5], **Nancy D. Lin**[5], **Benjamin D. Pollock**[6], **Jennifer L. Gatz**[3], **Shaun J. Grannis**[3,7], **Rohit Vashisht**[8], **Kanwal Ghauri**[1], **Camille Knepper**[6], **Sandy Leonard**[9], **Peter J. Embi**[10], **Garrett Jenkinson**[6], **Reyna Klesh**[9], **Omai B. Garner**[11], **Ayan Patel**[12], **Lisa Dahm**[12], **Aiden Barin**[12], **Dan M. Cooper**[12,13], **Tom Andriola**[12,14], **Carrie L. Byington**[12], **Bridgit O. Crews**[15], **Atul J. Butte**[8,12], **Jeff Allen**[16]

1 Reagan-Udall Foundation for the FDA, Washington, District of Columbia, United States of America, 2 Aetion, New York, New York, United States of America, 3 Regenstrief Institute, Indianapolis, Indiana, United States of America, 4 Optum Labs, Minnetonka, Minnesota, United States of America, 5 Health Catalyst, Salt Lake City, Utah, United States of America, 6 Mayo Clinic, Rochester, Minnesota, United States of America, 7 Department of Family Medicine, Indiana University School of Medicine, Indianapolis, Indiana, United States of America, 8 Bakar Computational Health Sciences Institute, University of California, San Francisco, San Francisco, California, United States of America, 9 HealthVerity, Philadelphia, Pennsylvania, United States of America, 10 Vanderbilt University Medical Center, Nashville, Tennessee, United States of America, 11 Department of Pathology and Laboratory Medicine, UCLA Medical Center, Los Angeles, California, United States of America, 12 Center for Data-Driven Insights and Innovation, University of California Health, Oakland, California, United States of America, 13 Pediatric Exercise and Genomics Research Center, University of California Irvine School of Medicine, Irvine, California, United States of America, 14 Office of Data and Information Technology, University of California, Irvine, Irvine, California, United States of America, 15 Department of Pathology and Laboratory Medicine, University of California, Irvine, Irvine, California, United States of America, 16 Friends of Cancer Research, Washington, District of Columbia, United States of America

☯ These authors contributed equally to this work.
* crodriguezwatson@reaganudall.org

**Data Availability Statement:** All relevant data are contained within the paper and its Supporting information files. Person-level data are unavailable.

## Abstract

### Background

Real-world performance of COVID-19 diagnostic tests under Emergency Use Authorization (EUA) must be assessed. We describe overall trends in the performance of serology tests in the context of real-world implementation.

### Methods

Six health systems estimated the odds of seropositivity and positive percent agreement (PPA) of serology test among people with confirmed SARS-CoV-2 infection by molecular test. In each dataset, we present the odds ratio and PPA, overall and by key clinical, demographic, and practice parameters.

### Results

A total of 15,615 people were observed to have at least one serology test 14–90 days after a positive molecular test for SARS-CoV-2. We observed higher PPA in Hispanic (PPA range:

**Funding:** Financial support for this work was provided in part by a grant from The Rockefeller Foundation (HTH 030 GA-S). BDP, CK, GJ used funding provided by Yale University-Mayo Clinic Center of Excellence in Regulatory Science and Innovation (CERSI), a joint effort between Yale University, Mayo Clinic, and the U.S. Food and Drug Administration (FDA) (3U01FD005938) (https://www.fda.gov/). AJB was funded by award number A128219 and Grant Number U01FD005978 from the FDA, which supports the UCSF-Stanford Center of Excellence in Regulatory Sciences and Innovation (CERSI). Its contents are solely the responsibility of the authors and do not necessarily represent the official views of the HHS or FDA. The funders had no role in study design, data collection and analysis, decision to publish, or preparation of the manuscript.

**Competing interests:** AJB is a co-founder and consultant to Personalis and NuMedii; consultant to Samsung, Mango Tree Corporation, and in the recent past, 10x Genomics, Helix, Pathway Genomics, and Verinata (Illumina); has served on paid advisory panels or boards for Geisinger Health, Regenstrief Institute, Gerson Lehman Group, AlphaSights, Covance, Novartis, Genentech, Merck, and Roche; is a shareholder in Personalis and NuMedii; is a minor shareholder in Apple, Facebook, Alphabet (Google), Microsoft, Amazon, Snap, Snowflake, 10x Genomics, Illumina, Nuna Health, Assay Depot (Scientist.com), Vet24seven, Regeneron, Sanofi, Royalty Pharma, Pfizer, BioNTech, AstraZeneca, Moderna, Biogen, Twist Bioscience, Pacific Biosciences, Editas Medicine, Invitae, Doximity, and Sutro, and several other non-health related companies and mutual funds; and has received honoraria and travel reimbursement for invited talks from Johnson and Johnson, Roche, Genentech, Pfizer, Merck, Lilly, Takeda, Varian, Mars, Siemens, Optum, Abbott, Celgene, AstraZeneca, AbbVie, Westat, several investment and venture capital firms, and many academic institutions, medical or disease specific foundations and associations, and health systems. AJB receives royalty payments through Stanford University, for several patents and other disclosures licensed to NuMedii and Personalis. AJB's research has been funded by NIH, Northrup Grumman (as the prime on an NIH contract), Genentech, Johnson and Johnson, FDA, Robert Wood Johnson Foundation, Leon Lowenstein Foundation, Intervalien Foundation, Priscilla Chan and Mark Zuckerberg, the Barbara and Gerson Bakar Foundation, and in the recent past, the March of Dimes, Juvenile Diabetes Research Foundation, California Governor's Office of Planning and Research, California Institute for

79–96%) compared to non-Hispanic (60–89%) patients; in those presenting with at least one COVID-19 related symptom (69–93%) as compared to no such symptoms (63–91%); and in inpatient (70–97%) and emergency department (93–99%) compared to outpatient (63–92%) settings across datasets. PPA was highest in those with diabetes (75–94%) and kidney disease (83–95%); and lowest in those with auto-immune conditions or who are immunocompromised (56–93%). The odds ratios (OR) for seropositivity were higher in Hispanics compared to non-Hispanics (OR range: 2.59–3.86), patients with diabetes (1.49–1.56), and obesity (1.63–2.23); and lower in those with immunocompromised or autoimmune conditions (0.25–0.70), as compared to those without those comorbidities. In a subset of three datasets with robust information on serology test name, seven tests were used, two of which were used in multiple settings and met the EUA requirement of PPA ≥87%. Tests performed similarly across datasets.

## Conclusion

Although the EUA requirement was not consistently met, more investigation is needed to understand how serology and molecular tests are used, including indication and protocol fidelity. Improved data interoperability of test and clinical/demographic data are needed to enable rapid assessment of the real-world performance of *in vitro* diagnostic tests.

## Introduction

Despite the availability of highly effective COVID-19 vaccines to prevent hospitalization and reduce mortality [1, 2], variants continue to fuel the surge of COVID-19 across the U.S. [3, 4]. High-quality diagnostic and serology tests are essential tools to better understand the epidemiology of COVID-19 and immunity after infection [5, 6]. Viruses and antibodies are primarily detectable within certain temporal windows [7–9]. However, many individuals infected with SARS-CoV-2 are asymptomatic or may not seek medical care because of mild symptoms [10]. In contrast to molecular diagnostic tests, serologic tests are informative even once the SARS-CoV-2 infection is no longer present [11, 12].

Currently, there are 90 authorized SARS-CoV-2 serology/antibody tests approved for Emergency Use Authorization (EUA) [13]. However, they have not undergone the same evidentiary review standards required for Food and Drug Administration (FDA) clearance due to the COVID-19 national emergency [14, 15]. There is a need to assess the real-world performance of these tests. Further, while large studies have shown that greater than 91% of people with active SARS-CoV-2 infection seroconvert [16, 17], the factors associated with seroconversion (e.g., pre-existing conditions, the severity of COVID-19 presentation) remain elusive.

From a public health perspective, confidence in the ability of serological tests to identify those with recent infections is critical for effective pandemic planning. Estimates of disease prevalence directly inform dynamic population estimates of susceptible, infected, and recovered, which are needed to understand the infectiousness of SARS-CoV-2 [18]. From a clinical perspective, an accurate understanding of SARS-CoV-2 exposure is necessary to understand disease presentation and a clinical course of action, especially when patients do not present with symptoms or present late in their disease course (e.g., post-acute sequelae of SARS-CoV-2). Additionally, identifying factors associated with seropositivity may elucidate potential mechanisms of action that may be foundational in the development of therapy and treatment plans.

Regenerative Medicine, L'Oreal, and Progenity. CLB has intellectual property in and receives royalties from BioFire, Inc. She serves as a scientific advisor to IDbyDNA (San Francisco, CA and Salt Lake City, UT); and is on the Board of the Commonwealth Fund. CK is a paid employee of Aetion and hold Aetion stock options. NES is an employee of Optum Labs and owns stock in the parent company UnitedHealth group. NDL was an employee of Health Catalyst at the time the work was performed. JLG is a full-time employee of Regenstrief Institute, which provides independent research services to entities including those within the pharmaceutical and medical device industries. SJG serves as Chief Medical Information Officer for the Indiana Health Information Exchange, and is a founding partner of Uppstroms, LLC. This does not alter our adherence to PLOS ONE policies on sharing data and materials.

To address these gaps, we characterize the performance of serology tests by estimating the positive percent agreement (PPA) of serological samples obtained from people known to be positive for SARS-CoV-2 infection by molecular assay (e.g., PCR). We also sought to identify factors associated with seropositivity. Findings from this study may facilitate understanding of the real-world performance of serology tests, many of which were issued under EUA, and may help inform our understanding of the immune response to SARS-CoV-2.

## Materials and methods

### Study population and setting

Six health systems (i.e., datasets) collaborated on the Diagnostics Evidence Accelerator (EA): Health Catalyst, Mayo Clinic, Optum Labs, Regenstrief Institute, the University of California Health System, and Aetion and HealthVerity. The EA is a consortium of leading experts in health systems research, regulatory science, data science, and epidemiology, specifically assembled to analyze health system data to address key questions related to COVID-19. The EA provides a platform for rapid learning and research using a common analytic plan. Health Catalyst, Mayo Clinic, and the University of California Health System all utilized electronic health records (EHR) data from their respective healthcare delivery systems. The Regenstrief Institute accessed EHR and public health data from the Indiana Health Information Exchange [19, 20], while Aetion sourced healthcare data from HealthVerity Marketplace encompassing medical claims, pharmacy claims, hospital chargemaster, and data collected directly from laboratories. Optum Labs data included de-identified medical, and pharmacy claims as well as laboratory results data utilized medical, and pharmacy claims from a single, large U.S. insurer as well as data directly from laboratories. We refer to these health systems as datasets A-F for the purposes of anonymity. Data sources included in the analysis are generally categorized as either payer (claims) or healthcare delivery systems. As illustrated in Fig 1, data were drawn from across the U.S. with heavy representation in California, Illinois, Ohio, and Michigan. Characteristics of participating data sources and representative populations are described in the S1 Table.

### Study design

In this retrospective cohort study, we identified patients across different settings (e.g., inpatient, outpatient, emergency department (ED), or long-term care facility) who tested positive

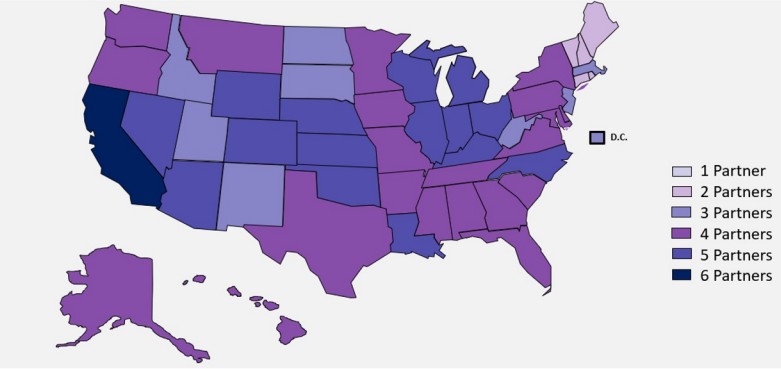

**Fig 1. Geographic coverage of data datasets.** Reprinted from brightcarbon.com under a CC BY license, with permission from Bright Carbon, original copyright (2021). Each color represents the number of data partners with a presence in each state but does not necessarily correspond to the number of people. The darkest color represents those where all six partners had a presence.

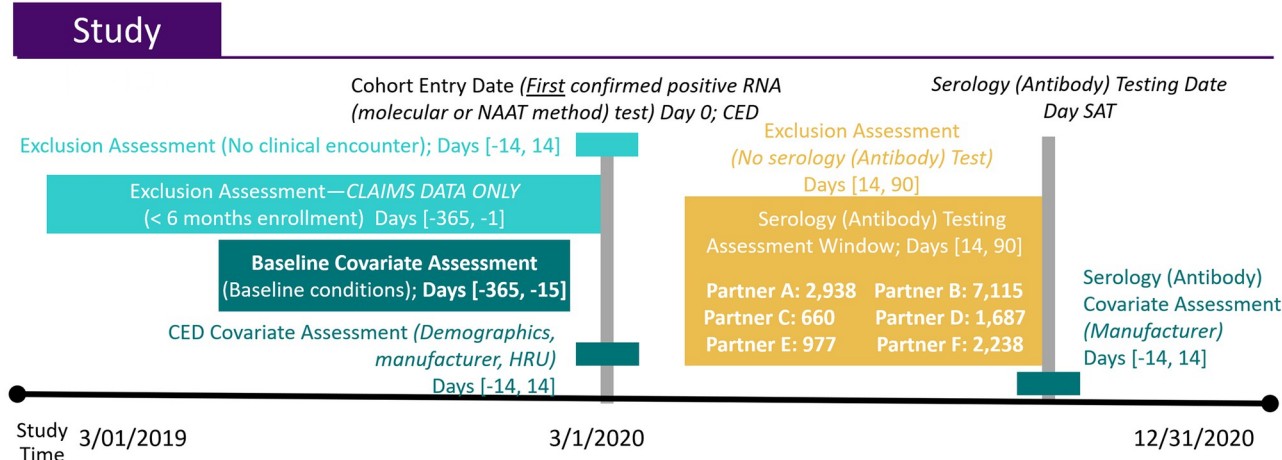

**Fig 2. Study design diagram.**

for SARS-CoV-2 ribonucleic acid (RNA) by molecular test between March–September 2020 and who received at least one subsequent serological test for SARS-CoV-2 immunoglobulin (Ig) G or Total antibody (Ab) from 14–90 days after the positive RNA test (Fig 2). We analyzed the first serology test in the 14–90-day follow-up period, which ended on December 31, 2020. "Date of RNA positive" served as the index (cohort entry) date and was defined hierarchically as either the date of 1) sample collection; 2) accession; or 3) result. Because the optimal time to observe a positive serology is at least two weeks after the index date, we only include patients who had at least one serology test 14–90 days after the index date [1–3, 7–9].

To minimize the effect of differential missingness between datasets, we applied the following rules: 1) included all persons with an office or telephone visit in the +/- 14 days around the index date to enable as complete an assessment of presenting symptoms as possible; 2) in claim systems, included only persons with at least six months of enrollment in the year before index; 3) estimated the proportion of patients at each site who had zero encounters in the prior year to contextualize our capture of pre-existing conditions, and 4) excluded variables from analysis if ≥30% of values were missing.

The Western—Copernicus Group (WCG) Institutional Review Board (IRB), the IRB of record for the Reagan-Udall Foundation for the FDA, reviewed the study and determined it to be non-human subjects research. Additionally, all legal and ethical approvals for use of the data included in this study were submitted, reviewed, and/or obtained locally at each contributing dataset by an IRB and/or governing board.

## Measures

**Outcomes.** The primary outcome of interest for the validation analysis was the PPA of positive antibody (IgG or total) from serology tests with positive RNA from molecular tests (e.g., PCR), which served as the reference standard. Serology tests reported in this analysis included: Abbott Architect IgG [21], Euroimmun IgG [22], Diazyme DZ-Lite SARS-CoV-2 IgG CLIA kit [23], Beckman SARS-CoV-2 IgG [24], Ortho Vitros IgG [25], Diasorin Liaison SARS-CoV-2 S1/S2 IgG [26], and Roche Elecsys Total Ab [27]. The Ortho Vitros was the only test used across multiple (3) datasets. We refer to these manufacturers—serological tests as Δ, Θ, Π, Λ, Ξ, Γ, and Ψ for anonymity. Molecular tests most reported in this analysis included: Hologic Panther Fusion [28], Hologic Aptima [29], Roche Cobas [30], Quest rRT-PCR [31],

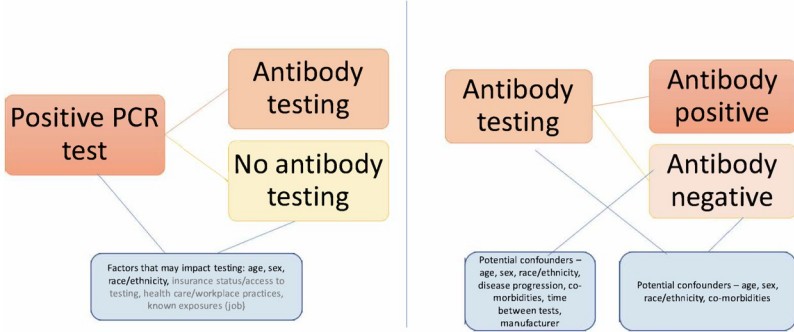

**Fig 3. Factors potentially associated with serological testing.** Pepe, 2001 Sep;2(3):249–60.

and Thermo Fisher Scientific Combo Kit [32]. We refer to these manufacturers—molecular tests as Σ, Φ, Ω, X, Y, and j for anonymity.

**Covariates.** We collected demographic, behavioral, and environmental characteristics, baseline clinical presentation, key comorbidities, and test characteristics, including manufacturer, according to a diagram illustrating potential factors associated with serology testing (Fig 3). We identified comorbidities and clinical presentation using phenotypes defined by the International Classification of Diseases 10 (ICD-10), and/or National Drug Codes. We identified comorbidities (pre-existing conditions) in the 365 days before the index date through 15 days before the index date. We provided coding algorithms for groups to use, while some groups used existing algorithms generated by their site. The ICD-10 codes used to identify comorbidities are listed in the S2 Table. We also stratified analyses by RNA tests conducted before June 15, 2020, which marked the beginning of the summer wave of infections in the first year of the pandemic, compared to on or after that date.

## Statistical analysis

Each contributing dataset ran its analysis according to a common protocol. Results were reviewed as a group to ensure alignment with the protocol and to review any protocol deviations. We calculated PPA as: *(Number of positive antibody results ÷ Number of positive RNA results) x 100*. We calculated PPA based on the first eligible serology test in the follow-up period overall and by age, sex, race, ethnicity, U.S. region, pregnancy status, pre-existing conditions, including but not limited to cardiovascular disease, obesity, hypertension, kidney disease, asthma, dementia, chronic liver disease, and smoking status. We also report the PPA by presenting symptoms, and serology tests at the time of the first serology test. We examined variations in PPA by serology tests and time, and serology tests and symptom presentation. We also examined variations in PPA by geography and care setting over time. We calculated exact (Clopper-Pearson) 95% confidence intervals (CI). We report significant differences where 95% CI have complete separation—although we did not conduct formal statistical comparisons of PPA between groups.

To study the odds of seropositivity, we estimated a model for the association to identify independent risk factors for seropositivity, assuming a binomial distribution for seropositivity status. Results are presented as the odds ratio (OR) and 95% CI that was calculated using score confidence intervals or exact CI [33]. All variables were treated as categorical. Symptoms were reported as a binary variable: "1" if any of the following symptoms were present: fever >100.4, abnormal chest imaging finding, high respiratory rate, low blood pressure, diarrhea, hypoglycemic, chest pain, delirium/confusion, headache, sore throat, cough, shortness of breath,

pneumonia, acute respiratory infection, acute respiratory distress, cardiovascular presentation, renal presentation; and "0" otherwise. For datasets with data covering >1 geographic catchment area, geography was included as either one of four U.S. Census regions, or nine U.S. Census divisions based on patient home zip code. Variables with >30% missing/unknown values were excluded from models (except for pregnancy, pre-existing condition, or presenting symptoms, all of which were included). Each dataset used automated backward selection to remove non-significant pre-existing conditions while forcing all other covariates into the model. All analyses were performed using SAS software, version 9.2 or higher (SAS Institute, North Carolina, U.S.); or the Aetion Evidence Platform v4.13 (including R v3.4.2), which includes audit trails of all transformations of raw data and a quality check of the data ingestion process.

## Results

Samples sizes across datasets ranged from 660–7,115; a total of 15,615 people with at least one serology test 14–90 days after the index date were included in the analyses. Between 35–65% of patients identified from health care delivery systems had no documented encounter in the system between 365 and 15 days before the index date. In contrast, only 11% of patients from national insurers reported having zero claims in the baseline period. As shown in Table 1, the serotested population was primarily 45–64 years of age (>40%), with a history of cardiovascular disease, including hypertension (8–70%). Race and ethnicity data were robust (<30% missing) in four datasets. The serotested population in those datasets was primarily White (>53%) and non-Hispanic (>65%), From datasets with national representation, persons from the Northeast (New England and Mid-Atlantic) were most represented in this serotested population. In datasets that represent regionally-based healthcare delivery systems, their population reflected their locations: Pacific and Midwest. Information on manufacturer test names was provided in four datasets. Generally, 2–3 primary tests were utilized in each dataset; 4 of 7 tests reported were used in >1 dataset. We did not observe any difference by age or sex in those for whom the test name was known versus unknown. In a single dataset with <30% of missing data on race/ethnicity, we observe over-representation of White and Hispanic people in those for whom the test name was known.

### Positive percent agreement (PPA) of serology among molecularly confirmed SARS-CoV-2

The overall PPA ranged from 65–90% across analytic datasets (Table 2). The real-world PPA met the EUA requirement of ≥87% in three datasets (A, B, D) [34]. Two of these datasets represented national administrative claims and associated results with the date the sample was collected or received by the laboratory; the third represented data from EHRs and associated results with the date the test was conducted, which is lagged further from the clinical interaction than the former. Overall PPA was likely influenced by the mix of serology tests represented in each dataset. Seven serological tests were reported in this analysis, of which two (Δ and Γ) met the EUA PPA requirements. Two tests were used across multiple datasets and performed similarly above the EUA requirement. PPA by serology test type varied across datasets; with three of five reporting significantly lower PPA from total antibody (PPA range: 69–90%) compared to IgG (PPA range: 87–92%); and two showing no difference. We observed no difference in PPA with antibody tests that target spike compared to nucleocapsid proteins.

PPA was significantly higher in Black (PPA range: 86–92%), as compared to White (PPA range: 78–86%), persons in at least two of the four datasets reporting robust race/ethnicity data. PPA was significantly higher in Hispanic (PPA range: 79–96%), compared to non-Hispanic (PPA range: 60–86%), patients. PPA appeared highest in those with diabetes (PPA

**Table 1. Clinical and demographic characteristics of patients with positive RNA and who underwent serological tests.**

| Datasets | | A | B | C | D | E | F |
|---|---|---|---|---|---|---|---|
| | | N = 2,938 (%) | N = 7,115 (%) | N = 660 (%) | N = 1,687 (%) | N = 977 (%) | N = 2,238 (%) |
| Age (years) | <20 | 118 (3.96) | 209 (2.94) | 16 (2.42) | 34 (2.02) | 20 (2.05) | 70 (3.13) |
| | 20–44 | 997 (33.42) | 2,381 (33.46) | 269 (40.76) | 496 (29.40) | 337 (34.49) | 682 (30.47) |
| | 45–54 | 698 (23.40) | 1,541 (21.66) | 105 (15.91) | 301 (17.84) | 171 (17.50) | 455 (20.33) |
| | 55–64 | 755 (25.31) | 1,560 (21.93) | 124 (18.79) | 365 (21.64) | 219 (22.42) | 484 (21.63) |
| | 65–74 | 288 (9.65) | 977 (13.73) | 102 (15.45) | 295 (17.49) | 142 (14.53) | 348 (15.55) |
| | 75–84 | 63 (2.11) | 359 (5.05) | 30 (4.55) | 152 (9.01) | 67 (6.86) | 152 (6.79) |
| | ≥85 | 19 (0.64) | 88 (1.24) | 14 (2.12) | 44 (2.61) | 21 (2.15) | 47 (2.10) |
| Sex | Female | 1,773 (59.44) | 3,890 (54.67) | 374 (56.67) | 949 (56.25) | 556 (56.91) | 1,350 (60.32) |
| | Male | 1,165 (39.05) | 3,225 (45.33) | 286 (43.33) | 738 (43.75) | 421 (43.09) | 888 (39.68) |
| Race | Black | NA[5] | 211 (2.97) | 19 (2.88) | 118 (6.99) | 66 (6.76) | 273 (12.20) |
| | White | NA | 1,268 (17.82) | 355 (53.79) | 1,321 (78.30) | 760 (77.79) | 1,784 (79.71) |
| | Asian | NA | 35 (0.49) | 67 (10.15) | 29 (1.72) | 44 (4.50) | 32 (1.43) |
| | Pacific islander/ native Hawaiian | NA | NA | NA | 11 (0.65) | 2 (0.20) | NA |
| | American Indian or Alaska native | NA | 1 (0.01) | NA | 41 (2.43) | 12 (1.23) | 11 (0.49) |
| | Other | NA | 866 (12.17) | 84 (12.73) | NA | NA | 6 (0.27) |
| | Unknown/ missing | 2,938 (100) | 4,734 (66.54) | 135 (20.45) | 167 (9.90) | 93 (9.52) | 132 (5.90) |
| Hispanic ethnicity | Yes | NA | 866 (12.17) | 178 (26.97) | 444 (26.32) | 124 (12.69) | 245 (10.95) |
| | No | NA | 1,515 (21.29) | 432 (65.45) | 1,212 (71.84) | 830 (84.95) | 1,867 (83.42) |
| | Unknown/ missing | 2,938 (100) | 4,734 (66.54) | 50 (7.58) | 31 (1.84) | 23 (2.35) | 126 (5.63) |
| Pre-existing conditions[1,2] | Diabetes | 634 (21.25) | 1,215 (17.08) | 102 (15.45) | 307 (18.20) | NA | 213 (9.52) |
| | Cardiovascular disease | 1,332 (44.65) | 2,974 (41.80) | 307 (46.52) | 639 (37.88) | NA | 116 (5.18) |
| | Hypertension | 1,096 (36.74) | 2,494 (35.05) | 192 (29.09) | 532 (31.54) | NA | 63 (2.82) |
| | Immunocompromised (e.g., HIV, cancer) or auto-immune disorder | 349 (11.70)[7] | 708 (9.05) | 121 (18.33) | 110 (6.52) | NA | 114 (5.09) |
| | Asthma | 334 (11.20) | 575 (8.08) | 45 (6.82) | 131 (7.77) | NA | 133 (5.94) |
| | Kidney disease | 141 (4.73) | 317 (4.46) | 118 (17.88) | 195 (11.56) | NA | 90 (4.02) |
| | Chronic lung conditions | 443 (14.85) | 878 (12.34) | NA | 208 (12.33) | NA | 59 (2.64) |
| | Any liver disease | 227 (7.61) | 391 (5.50) | 60 (9.09) | 81 (4.80) | NA | 15 (0.67) |
| | Obesity | 829 (27.79) | 655 (9.21) | 83 (12.58) | 250 (14.82) | NA | 169 (7.55) |
| | Dementia | 23 (0.77) | NA | 8 (1.21) | 13 (0.77) | NA | 15 (0.67) |
| | None of above comorbidities | 1,033 (34.63) | 3,230 (45.40) | NA | 870 (51.57) | NA | 1,718 (76.76) |

(*Continued*)

**Table 1.** (Continued)

| Datasets | | A | B | C | D | E | F |
|---|---|---|---|---|---|---|---|
| | | N = 2,938 (%) | N = 7,115 (%) | N = 660 (%) | N = 1,687 (%) | N = 977 (%) | N = 2,238 (%) |
| Pregnancy status[1,3] | Yes | 82 (2.75) | NA | 40 (6.06) | NA | NA | NA |
| | No | 1,688 (56.59) | NA | 334 (50.61) | NA | NA | NA |
| | Unknown/ missing | NA | 7,115 (100) | NA | NA | NA | NA |
| Geographic divisions and regions[4] | New England | 43 (1.44) | 2,669 (37.51) | NA | NA | 0 (0) | NA |
| | Mid-Atlantic | 1,724 (57.79) | NA | NA | NA | 2 (0.20) | NA |
| | South Atlantic | 333 (11.16) | 2,952 (41.49) | NA | NA | 79 (8.09) | NA |
| | East south central | 16 (0.54) | NA | NA | NA | 61 (6.24) | NA |
| | West south central | 215 (7.21) | NA | NA | NA | 1 (0.10) | NA |
| | East north central | 154 (5.16) | 293 (4.12) | NA | NA | 448 (45.85) | 2,238 (100) |
| | West north central | 22 (0.74) | NA | NA | NA | 3 (0.31) | NA |
| | Mountain | 23 (0.77) | 1,201 (16.88) | NA | NA | 374 (38.28) | NA |
| | Pacific | 210 (7.04) | NA | 660 (100) | NA | 8 (0.82) | NA |
| | Unknown/ missing | 198 (6.64) | NA | NA | 1,687(100) | 1 (0.10) | NA |
| Presenting symptoms[1] | No presenting symptoms identified | 1,874 (62.82) | 4,193 (58.93) | 404 (61.21) | 917 (54.36) | NA | NA |
| | Fever >100.4 | 80 (2.68) | 265 (3.72) | NA | NA | NA | NA |
| | Low blood pressure | 10 (0.34) | NA | NA | NA | NA | NA |
| | Diarrhea | 32 (1.07) | 79 (1.11) | 31 (4.70) | 47 (2.79) | NA | NA |
| | Hypoglycemic | 7 (0.23) | NA | NA | NA | NA | NA |
| | Chest pain | 120 (4.02) | 298 (4.19) | 43 (6.52) | 68 (4.03) | NA | NA |
| | Delirium/confusion | 67 (2.25) | 24 (0.34) | NA | 126 (7.47) | NA | NA |
| | Headache | 69 (2.31) | 146 (2.05) | 20 (3.03) | 23 (1.36) | NA | NA |
| | Sore throat | 38 (1.27) | 95 (1.34) | NA | 17 (1.01) | NA | NA |
| | Cough | 266 (8.92) | 810 (11.38) | 100 (15.15) | 68 (4.03) | NA | NA |
| | Shortness of breath | 255 (8.55) | 538 (7.56) | 78 (11.82) | 166 (9.84) | NA | NA |
| | Pneumonia | 165 (5.53) | 450 (6.32) | 78 (11.82) | 337 (19.98) | NA | NA |
| | Acute respiratory infection | 62 (2.08) | 22 (0.31) | 22 (3.33) | 298 (17.66) | NA | NA |
| | Acute respiratory distress, arrest, or failure | 53 (1.78) | 292 (4.10) | 43 (6.52) | 10 (0.59) | NA | NA |
| | Cardiovascular condition | 609 (20.42) | 1,719 (24.16) | 131 (19.58) | 598 (34.45) | NA | NA |
| | Renal condition | 61 (2.04) | 214 (3.01) | 57 (8.64) | NA | NA | NA |
| | ≥ 1 symptom above | 1,064 (35.67) | 2,922 (41.07) | 256 (38.79) | 770 (45.64) | NA | NA |
| Serological test type | IgG | 2,769 (92.83) | 6397 (89.91) | 660 (100) | 1,617 (95.85) | 593 (60.70) | 1,911 (85.39) |
| | Total antibody | 169 (5.67) | 718 (10.09) | NA | 42 (2.49) | 384 (39.30) | 327 (14.61) |
| | Unknown/ missing | NA | NA | NA | 28 (1.66) | NA | NA |

(*Continued*)

**Table 1.** (Continued)

| Datasets | | A | B | C | D | E | F |
|---|---|---|---|---|---|---|---|
| | | N = 2,938 (%) | N = 7,115 (%) | N = 660 (%) | N = 1,687 (%) | N = 977 (%) | N = 2,238 (%) |
| Manufacturer—serological test name | Δ | 1,604 (53.77) | 983 (13.82) | NA | NA | NA | NA |
| | Θ | NA | NA | 43 (6.52) | NA | NA | NA |
| | Π | 2 (0.07) | NA | NA | NA | 314 (32.14) | NA |
| | Λ | 4 (0.13) | NA | 290 (43.94) | NA | NA | NA |
| | Ξ | NA | NA | 60 (9.09) | NA | NA | NA |
| | Γ | 637 (21.35) | 513 (7.21) | NA | NA | 279 (28.56) | NA |
| | Ψ | NA | NA | NA | NA | 384 (39.90) | NA |
| | Unknown/ missing | 691 (23.16) | 5,619 (78.97) | 267 (40.45) | 1,687 (100) | NA | 2,238 (100) |
| Manufacturer—molecular test name | Y | NA | 44 (0.62) | NA | NA | NA | NA |
| | X | 267 (8.95) | 403 (5.66) | NA | NA | NA | NA |
| | Σ | 272 (9.12) | 367 (5.16) | NA | NA | NA | NA |
| | Φ | NA | 85 (1.19) | NA | NA | NA | NA |
| | Ω 6 | 71 (2.38) | NA | NA | NA | NA | NA |
| | j | 18 (0.60) | NA | NA | NA | NA | NA |
| | Unknown/missing | 2,310 (77.44) | 6,216 (87.36) | NA | NA | NA | NA |
| Care setting (where RNA test occurred) | Inpatient | 86 (2.88) | 151 (2.12) | 97 (14.70) | NA | 53 (5.42) | NA |
| | Outpatient | 1,407 (47.17) | 6,685 (93.96) | 563 (85.30) | NA | 777 (79.53) | NA |
| | ED | 143 (4.79) | 279 (3.92) | NA | NA | 147 (15.05) | NA |
| | Unknown/ missing | 1,302 (43.65) | NA | NA | NA | NA | NA |
| Calendar time (based on RNA test) | Before June 15, 2020 | 2,149 (72.04) | 3,761 (52.86) | 275 (41.67) | 476 (28.22) | 472 (48.31) | 664 (29.67) |
| | On or after June 15, 2020 | 789 (26.45) | 3,354 (47.14) | 385 (58.33) | 1,211 (71.78) | 505 (51.69) | 1,574 (70.33) |
| Smoking status | Has a history of smoking | NA | NA | NA | 256 (15.17) | NA | NA |
| | No history | NA | NA | NA | 1,431 (84.83) | NA | NA |

[1.] Phenotypes (code-sets) of ICD-10, medication, and LOINC are provided in the S2 Table. Conditions may be identified using ICD-10, medication, or both.

[2.] Pre-existing conditions were assessed 365 days before the index date and were not mutually exclusive.

[3.] Pregnancy Status was assessed up to 40 weeks before the index date (among women only).

[4.] Geographic regions were based on patient home zip code and defined by the U.S. Census Bureau (https://www2.census.gov/geo/pdfs/maps-data/maps/reference/us_regdiv.pdf) and mapped by census track zip code. States included in each region are as follows: **New England**: Connecticut, Maine, Massachusetts, New Hampshire, Rhode Island, Vermont; **Mid Atlantic**: New Jersey, New York, Pennsylvania; **East North Central**: Indiana, Illinois, Michigan, Ohio, Wisconsin; **West North Central**: Iowa, Nebraska, Kansas, North Dakota, Minnesota, South Dakota, Missouri; **South Atlantic**: Delaware, District of Columbia, Florida, Georgia, Maryland, North Carolina, South Carolina, Virginia, West Virginia; **East South Central**: Alabama, Kentucky, Mississippi, Tennessee; **West South Central**: Arkansas, Louisiana, Oklahoma, Texas; **Mountain:** Arizona, Colorado, Idaho, New Mexico, Montana, Utah, Nevada, Wyoming; **Pacific:** Alaska, California, Hawaii, Oregon, Washington.

[5.] Data were not available.

[6.] Ω not specified and may not have received an EUA.

[7.] Dataset A is only looking at the autoimmune diseases

**Table 2. Positive percent agreement (PPA) 14–90 days after positive RNA test.**

| Datasets | | A | B | C | D | E | F |
|---|---|---|---|---|---|---|---|
| | | N = 2,938 | N = 7,115 | N = 660 | N = 1,687 | N = 977 | N = 2,238 |
| | | PPA (95% confidence interval) | | | | | |
| Overall | | 0.91 (0.90, 0.92) | 0.90 (0.89, 0.90) | 0.65 (0.62, 0.69) | 0.88 (0.86, 0.89) | 0.84 (0.82, 0.86) | 0.79 (0.78, 0.81) |
| Age (Years) | <20 | 0.92 (0.85, 0.96) | 0.85 (0.79, 0.89) | 0.81 (0.54, 0.96) | 0.91 (0.82, 1.00) | 0.8 (0.58, 0.92) | 0.77 (0.66, 0.86) |
| | 20–44 | 0.90 (0.88, 0.92) | 0.87 (0.86, 0.88) | 0.62 (0.56, 0.68) | 0.85 (0.82, 0.88) | 0.87 (0.83, 0.90) | 0.75 (0.72, 0.78) |
| | 45–54 | 0.93 (0.91, 0.95) | 0.9 (0.89, 0.92) | 0.67 (0.57, 0.76) | 0.88 (0.85, 0.92) | 0.89 (0.84, 0.93) | 0.81 (0.77, 0.84) |
| | 55–64 | 0.91 (0.89, 0.93) | 0.91 (0.90, 0.92) | 0.69 (0.60, 0.77) | 0.88 (0.84, 0.91) | 0.82 (0.77, 0.87) | 0.79 (0.75, 0.82) |
| | 65–74 | 0.93 (0.89, 0.95) | 0.92 (0.90, 0.94) | 0.65 (0.55, 0.74) | 0.92 (0.88, 0.95) | 0.79 (0.71, 0.85) | 0.83 (0.79, 0.87) |
| | 75–84 | 0.97 (0.89, 1.00) | 0.92 (0.89, 0.95) | 0.67 (0.47, 0.83) | 0.86 (0.81, 0.92) | 0.82 (0.71, 0.89) | 0.86 (0.79, 0.91) |
| | ≥85 | 0.89 (0.67, 0.99) | 0.91 (0.83, 0.96) | 0.71 (0.42, 0.92) | 0.86 (0.76, 0.97) | 0.71 (0.50, 0.86) | 0.87 (0.74, 0.95) |
| Sex | Female | 0.91 (0.90, 0.93) | 0.89 (0.88, 0.90) | 0.61 (0.56, 0.66) | 0.87 (0.84, 0.89) | 0.85 (0.82, 0.88) | 0.77 (0.75, 0.80) |
| | Male | 0.91 (0.90, 0.93) | 0.90 (0.89, 0.91) | 0.71 (0.66, 0.76) | 0.89 (0.87, 0.91) | 0.83 (0.79, 0.86) | 0.82 (0.80, 0.85) |
| Race | Black | NA[5] | 0.95 (0.91, 0.98) | 0.68 (0.43, 0.87) | 0.92 (0.88, 0.97) | 0.86 (0.76, 0.93) | 0.86 (0.81, 0.90) |
| | White | NA | 0.88 (0.86, 0.90) | 0.60 (0.55, 0.65) | 0.86 (0.85, 0.88) | 0.83 (0.80, 0.85) | 0.78 (0.76, 0.80) |
| | Asian | NA | 0.94 (0.81, 0.99) | 0.67 (0.55, 0.78) | 0.90 (0.79, 1.00) | 0.84 (0.71, 0.92) | 0.75 (0.57, 0.89) |
| | Pacific Islander/ Native Hawaiian | NA | NA | NA | 1.00 (1.00, 1.00) | 1.00 (0.34, 1.00) | NA |
| | American Indian or Alaska Native | NA | 1.00 (0.05, 1.00) | NA | 1.00 (1.00, 1.00) | 0.75 (0.47, 0.91) | 1.00 (0.72, 1.00) |
| | Other | NA | 0.89 (0.88, 0.90) | 0.69 (0.58, 0.79) | NA | NA | 1.00 (0.54, 1.00) |
| | Unknown/ Missing | NA | 0.89 (0.88, 0.90) | 0.76 (0.68, 0.83) | 0.9 (0.86, 0.95) | 0.96 (0.89, 0.99) | 0.87 (0.80, 0.92) |
| Hispanic Ethnicity | Yes | NA | 0.94 (0.93, 0.96) | 0.79 (0.73, 0.85) | 0.94 (0.91, 0.96) | 0.96 (0.91, 0.98) | 0.92 (0.88, 0.95) |
| | No | NA | 0.89 (0.87, 0.91) | 0.60 (0.55, 0.65) | 0.86 (0.84, 0.88) | 0.82 (0.80, 0.85) | 0.78 (0.76, 0.79) |
| | Unknown/ Missing | NA | 0.89 (0.88, 0.90) | 0.64 (0.49, 0.77) | 0.81 (0.67, 0.95) | 0.87 (0.68, 0.95) | 0.8 (0.72, 0.87) |

*(Continued)*

**Table 2.** (Continued)

| Datasets | | A | B | C | D | E | F |
|---|---|---|---|---|---|---|---|
| | | N = 2,938 | N = 7,115 | N = 660 | N = 1,687 | N = 977 | N = 2,238 |
| Pre-existing Conditions[1,2] | Diabetes | 0.94 (0.92, 0.96) | 0.94 (0.92, 0.95) | 0.75 (0.66, 0.83) | 0.91 (0.88, 0.94) | NA | 0.85 (0.79, 0.89) |
| | Cardiovascular disease | 0.92 (0.90, 0.93) | 0.92 (0.91, 0.93) | 0.67 (0.62, 0.72) | 0.87 (0.84, 0.89) | NA | 0.83 (0.75, 0.89) |
| | Hypertension | 0.93 (0.91, 0.94) | 0.92 (0.91, 0.93) | 0.71 (0.64, 0.77) | 0.89 (0.86, 0.91) | NA | 0.87 (0.77, 0.94) |
| | Immunocompromised (e.g., HIV, Cancer,) or Auto-immune disorders | 0.93 (0.97, 1.00)[6] | 0.88 (0.85, 0.90) | 0.56 (0.47, 0.65) | 0.70 (0.61, 0.79) | NA | 0.73 (0.64, 0.81) |
| | Asthma | 0.89 (0.85, 0.92) | 0.9 (0.87, 0.92) | 0.62 (0.47, 0.76) | 0.82 (0.76, 0.89) | NA | 0.74 (0.65, 0.81) |
| | Kidney Disease | 0.92 (0.86, 0.96) | 0.95 (0.92, 0.97) | 0.75 (0.67, 0.83) | 0.9 (0.86, 0.94) | NA | 0.83 (0.74, 0.90) |
| | Chronic Lung conditions | 0.90 (0.86, 0.92) | 0.90 (0.88, 0.92) | NA | 0.86 (0.81, 0.90) | NA | 0.75 (0.62, 0.85) |
| | Any liver disease | 0.93 (0.89, 0.96) | 0.90 (0.87, 0.93) | 0.65 (0.52, 0.77) | 0.88 (0.80, 0.95) | NA | 0.80 (0.52, 0.96) |
| | Obesity | 0.93 (0.92, 0.95) | 0.92 (0.89, 0.94) | 0.81 (0.71, 0.89) | 0.88 (0.84, 0.92) | NA | 0.83 (0.76, 0.88) |
| | Dementia | 1.00 (0.85, 1.00) | NA | 0.75 (0.35, 0.97) | 1.00 (1.00, 1.00) | NA | 0.80 (0.77, 0.81) |
| | None of above comorbidities | 0.91 (0.89, 0.93) | 0.88 (0.87, 0.89) | NA | 0.89 (0.87, 0.91) | NA | 0.79 (0.52, 0.96) |
| Pregnancy Status[1,3] | Yes | 0.89 (0.80, 0.95) | NA | 0.58 (0.41, 0.73) | 0.87 (0.79, 0.96) | NA | NA |
| | No | 0.92 (0.90, 0.93) | NA | 0.61 (0.56, 0.67) | 0.86 (0.84, 0.89) | NA | NA |
| | Unknown/ Missing | NA | NA | NA | NA | NA | NA |
| Geographic Divisions and Regions [4] | New England | 0.77 (0.61, 0.88) | 0.9 (0.89, 0.91) | NA | NA | NA | NA |
| | Mid-Atlantic | 0.92 (0.91, 0.94) | NA | NA | NA | 1.00 (0.94, 1.00) | NA |
| | South Atlantic | 0.90 (0.86, 0.93) | 0.89 (0.88, 0.90) | NA | NA | 0.84 (0.74, 0.90) | NA |
| | East South Central | 0.81 (0.54, 0.96) | NA | NA | NA | 0.87 (0.76, 0.93) | NA |
| | West South Central | 0.93 (0.89, 0.96) | NA | NA | NA | 1.00 (0.21, 1.00) | NA |
| | East North Central | 0.86 (0.80, 0.91) | 0.83 (0.79, 0.87) | NA | NA | 0.79 (0.75, 0.82) | 0.79 (0.78, 0.81) |
| | West North Central | 0.82 (0.60, 0.95) | NA | NA | NA | 1.00 (0.44, 1.00) | NA |
| | Mountain | 0.96 (0.78, 1.00) | 0.91 (0.89, 0.92) | NA | NA | 0.91 (0.88, 0.94) | NA |
| | Pacific | 0.90 (0.85, 0.94) | NA | 0.65 (0.62, 0.69) | NA | 0.50 (0.22, 0.78) | NA |
| | Unknown/ Missing | 0.94 (0.90, 0.97) | NA | NA | 0.88 (0.86, 0.89) | 1.00 (0.21, 1.00) | NA |

(*Continued*)

**Table 2.** (*Continued*)

| Datasets | | A | B | C | D | E | F |
|---|---|---|---|---|---|---|---|
| | | N = 2,938 | N = 7,115 | N = 660 | N = 1,687 | N = 977 | N = 2,238 |
| Presenting Symptoms[1] | No presenting symptoms identified | 0.91 (0.89, 0.92) | 0.88 (0.87, 0.89) | 0.63 (0.58, 0.68) | 0.86 (0.84, 0.88) | NA | NA |
| | ≥ 1 symptom below | 0.93 (0.91, 0.94) | 0.92 (0.91, 0.93) | 0.69 (0.63, 0.75) | 0.89 (0.87, 0.92) | NA | NA |
| | Fever >100.4 | 0.95 (0.88, 0.99) | 0.91 (0.86, 0.94) | NA | NA | NA | NA |
| | Low blood pressure | 0.90 (0.55, 1.00) | NA | NA | NA | NA | NA |
| | Diarrhea | 0.88 (0.71, 0.96) | 0.96 (0.89, 0.99) | 0.61 (0.42, 0.78) | 0.85 (0.75, 0.95) | NA | NA |
| | Hypoglycemic | 0.71 (0.29, 0.96) | NA | NA | NA | NA | NA |
| | Chest pain | 0.93 (0.86, 0.97) | 0.90 (0.86, 0.93) | 0.74 (0.59, 0.86) | 0.84 (0.75, 0.93) | NA | NA |
| | Delirium/Confusion | 0.93 (0.83, 0.98) | 0.92 (0.73, 0.99) | NA | 0.96 (0.93, 0.99) | NA | NA |
| | Headache | 0.88 (0.78, 0.95) | 0.89 (0.83, 0.94) | 0.65 (0.41, 0.85) | 0.83 (0.67, 0.98) | NA | NA |
| | Sore throat | 0.92 (0.79, 0.98) | 0.85 (0.77, 0.92) | NA | 0.82 (0.64, 1.00) | NA | NA |
| | Cough | 0.93 (0.89, 0.96) | 0.92 (0.89, 0.93) | 0.74 (0.64, 0.82) | 0.78 (0.68, 0.88) | NA | NA |
| | Shortness of breath | 0.94 (0.90, 0.96) | 0.94 (0.91, 0.96) | 0.73 (0.62, 0.82) | 0.86 (0.80, 0.91) | NA | NA |
| | Pneumonia | 0.96 (0.91, 0.98) | 0.97 (0.95, 0.98) | 0.82 (0.72, 0.90) | 0.93 (0.91, 0.96) | NA | NA |
| | Acute respiratory infection | 0.92 (0.82, 0.97) | 0.86 (0.65, 0.97) | 0.68 (0.45, 0.86) | 0.95 (0.93, 0.98) | NA | NA |
| | Acute respiratory distress, arrest, or failure | 0.96 (0.87, 1.00) | 0.91 (0.88, 0.94) | 0.84 (0.69, 0.93) | 0.8 (0.55, 1.00) | NA | NA |
| | Cardiovascular condition | 0.94 (0.92, 0.96) | 0.92 (0.91, 0.94) | 0.73 (0.64, 0.80) | 0.89 (0.87, 0.92) | NA | NA |
| | Renal Condition | 0.93 (0.84, 0.98) | 0.93 (0.89, 0.96) | 0.79 (0.66, 0.89) | NA | NA | NA |
| Serological Test Type | IgG | 0.92 (0.91, 0.93) | 0.9 (0.89, 0.90) | 0.65 (0.62, 0.69) | 0.88 (0.86, 0.90) | 0.87 (0.84, 0.90) | 0.79 (0.77, 0.80) |
| | Total Antibody | 0.87 (0.81, 0.92) | 0.9 (0.87, 0.92) | NA | 0.69 (0.55, 0.83) | 0.8 (0.75, 0.83) | 0.83 (0.79, 0.87) |
| | Unknown/ Missing | NA | NA | NA | 0.96 (0.90, 1.00) | NA | NA |

(*Continued*)

**Table 2.** (Continued)

| Datasets | | A | B | C | D | E | F |
|---|---|---|---|---|---|---|---|
| | | N = 2,938 | N = 7,115 | N = 660 | N = 1,687 | N = 977 | N = 2,238 |
| Manufacturer—serological test name[7] | Δ | 0.91 (0.89, 0.92) | 0.89 (0.87, 0.91) | NA | NA | NA | NA |
| | Θ | NA | NA | 0.81 (0.67, 0.92) | NA | NA | NA |
| | Π | 0.50 (0.01, 0.99) | NA | NA | NA | 0.82 (0.78, 0.86) | NA |
| | Λ | 1 (0.40, 1.00) | NA | 0.70 (0.65, 0.76) | NA | NA | NA |
| | Ξ | NA | NA | 0.72 (0.59, 0.83) | NA | NA | NA |
| | Γ | 0.92 (0.90, 0.94) | 0.91 (0.88, 0.93) | NA | NA | 0.92 (0.89, 0.95) | NA |
| | Ψ | NA | NA | NA | NA | 0.8 (0.75, 0.83) | NA |
| | Unknown/ Missing | 0.93 (0.91, 0.95) | 0.9 (0.89, 0.90) | 0.56 (0.50, 0.62) | 0.88 (0.86, 0.89) | NA | 0.79 (0.78, 0.81) |
| Manufacturer—molecular test name | Y | NA | 0.91 (0.78, 0.97) | NA | NA | NA | NA |
| | X | 0.90 (0.85, 0.93) | 0.84 (0.80, 0.87) | NA | NA | NA | NA |
| | Σ | 0.94 (0.90, 0.96) | 0.92 (0.89, 0.95) | NA | NA | NA | NA |
| | Φ | NA | 0.91 (0.82, 0.96) | NA | NA | NA | NA |
| | Ω | 0.94 (0.86, 0.98) | NA | NA | NA | NA | NA |
| | j | 0.83 (0.59, 0.96) | NA | NA | NA | NA | NA |
| | Unknown/Missing | 0.91 (0.90, 0.93) | 0.90 (0.89, 0.91) | NA | NA | NA | NA |
| Care Setting (where RNA test occurred) | Inpatient | 0.97 (0.90, 0.99) | 0.97 (0.93, 0.99) | 0.77 (0.68, 0.85) | NA | 0.7 (0.56, 0.80) | NA |
| | Outpatient | 0.92 (0.91, 0.93) | 0.89 (0.88, 0.90) | 0.63 (0.59, 0.67) | NA | 0.84 (0.81, 0.86) | NA |
| | ED | 0.99 (0.95, 1.00) | 0.96 (0.93, 0.98) | NA | NA | 0.93 (0.88, 0.96) | NA |
| | Unknown/ Missing | 0.88 (0.88, 0.91) | NA | NA | NA | NA | NA |
| Calendar Time (based on RNA test) | Before June 15, 2020 | 0.92 (0.91, 0.93) | 0.92 (0.91, 0.93) | 0.61 (0.55, 0.67) | 0.92 (0.90, 0.95) | 0.84 (0.80, 0.87) | 0.80 (0.77, 0.83) |
| | On or after June 15, 2020 | 0.90 (0.88, 0.92) | 0.87 (0.86, 0.98) | 0.68 (0.63, 0.73) | 0.86 (0.84, 0.88) | 0.85 (0.81, 0.88) | 0.79 (0.77, 0.81) |

(*Continued*)

**Table 2.** (Continued)

| Datasets | | A | B | C | D | E | F |
|---|---|---|---|---|---|---|---|
| | | N = 2,938 | N = 7,115 | N = 660 | N = 1,687 | N = 977 | N = 2,238 |
| Smoking Status | Has History of Smoking | NA | NA | NA | 0.86 (0.82, 0.91) | NA | NA |
| | No History | NA | NA | NA | 0.88 (0.86, 0.90) | NA | NA |

[1.] Phenotypes (code-sets) of ICD-10, medication, and LOINC are provided in the S2 Table. Conditions may be identified using ICD-10, medication, or both.

[2.] Pre-existing conditions were assessed 365 days before the index date and were not mutually exclusive.

[3.] Pregnancy Status was assessed up to 40 weeks before the index date.

[4.] Geographic regions were based on patients' home codes code and defined by the U.S. Census Bureau (https://www2.census.gov/geo/pdfs/maps-data/maps/reference/us_regdiv.pdf) and mapped by census track zip code. States included in each region are as follows: **New England**: Connecticut, Maine, Massachusetts, New Hampshire, Rhode Island, Vermont; **Mid Atlantic**: New Jersey, New York, Pennsylvania; **East North Central**: Indiana, Illinois, Michigan, Ohio, Wisconsin; **West North Central**: Iowa, Nebraska, Kansas, North Dakota, Minnesota, South Dakota, Missouri; **South Atlantic**: Delaware, District of Columbia, Florida, Georgia, Maryland, North Carolina, South Carolina, Virginia, West Virginia; **East South Central**: Alabama, Kentucky, Mississippi, Tennessee; **West South Central**: Arkansas, Louisiana, Oklahoma, Texas; **Mountain:** Arizona, Colorado, Idaho, New Mexico, Montana, Utah, Nevada, Wyoming; **Pacific:** Alaska, California, Hawaii, Oregon, Washington.

[5.] Data was not available

[6.] Dataset A is only looking at the autoimmune diseases

[7.] Shaded cells represent a small sample size (n<40) or non-robust data capture (>30% missing).

range: 75–94%) and kidney disease (PPA range: 75–95%), and lowest in those with conditions that leave them immunocompromised (PPA range: 56–93%). We observed higher PPA in the inpatient (PPA range: 70–97%) or ED (PPA range: 93–99%) setting compared to outpatient (PPA range: 63–92%). There was some evidence of higher PPA among patients with at least one COVID-19 related symptoms as compared to those with none (PPA range: 63–91%) among two datasets (B and D); and was particularly high for select conditions like pneumonia (PPA range: 82–97%).

However, differences in the PPA by the presence of symptoms do not appear to be explained by the test. A stratified analysis by test comparing those with and without symptoms (Table 3) showed no significant difference in PPA. PPA trends by calendar time were not consistent across datasets.

### Factors associated with seropositivity

In adjusted models (Figs 4–9), the OR for seropositivity was significantly elevated in Hispanic compared to non-Hispanic ethnicity (OR range: 2.59–3.86); among those with pre-existing diabetes (OR range: 1.49–1.56) and obesity (1.63–2.23) as compared to those without pre-existing conditions; and among those observed in the ED compared to outpatient (OR range: 2.49–10.97). The OR for seropositivity was significantly lower in those with pre-existing immunocompromised or autoimmune conditions compared to those without such conditions (OR range: 0.25–0.70). In two of three datasets that included pre-existing cardiovascular disease in the OR model, the OR for seropositivity was significantly lower in persons with, compared to those without, such conditions (OR range: 0.49–0.57). The OR for seropositivity tended to be lower on or after June 15 compared to prior in half the datasets, but differences were not significant in the other half.

### Discussion

Serology tests are an important instrument in the toolkit to understand the epidemiology of COVID-19 because of their ability to identify persons with prior infection who may present

**Table 3. Positive percent agreement (PPA) by serology tests and presence of symptoms.**

| Manufacturer—serological test name Datasets | | A | B | C | D | E | F |
|---|---|---|---|---|---|---|---|
| | | N = 2,938 | N = 7,115 | N = 660 | N = 1,687 | N = 977 | N = 2,238 |
| | | PPA (95% confidence interval) | | | | | |
| Δ | No symptoms identified | 0.90 (0.88, 0.92) | 0.89 (0.86, 0.91) | NA | NA | NA | NA |
| | ≥ 1 symptom identified | 0.92 (0.89, 0.94) | 0.89 (0.85, 0.92) | NA | NA | NA | NA |
| Θ | No symptoms identified | NA[1] | NA | 0.69 (0.39, 0.91) | NA | NA | NA |
| | ≥ 1 symptom identified | NA | NA | 0.87 (0.69, 0.96) | NA | NA | NA |
| Π | No symptoms identified | 0.00 (0.00, 0.98) | NA | NA | NA | NA | NA |
| | ≥ 1 symptom identified | 1.00 (0.03, 1.00) | NA | NA | NA | NA | NA |
| Λ | No symptoms identified | 1.00 (0.4, 1.00) | NA | 0.69 (0.61, 0.75) | NA | NA | NA |
| | ≥ 1 symptom identified | 0.00 (0.00, 1.00) | NA | 0.73 (0.64, 0.81) | NA | NA | NA |
| Ξ | No symptoms identified | NA | NA | 0.69 (0.49, 0.85) | NA | NA | NA |
| | ≥ 1 symptom identified | NA | NA | 0.74 (0.55, 0.88) | NA | NA | NA |
| Γ | No symptoms identified | 0.92 (0.89, 0.94) | 0.88 (0.84, 0.92) | NA | NA | NA | NA |
| | ≥ 1 symptom identified | 0.93 (0.89, 0.96) | 0.95 (0.91, 0.98) | NA | NA | NA | NA |
| Unknown/Missing | No symptoms identified | 0.92 (0.88, 0.94) | 0.88 (0.87, 0.89) | 0.56 (0.49, 0.64) | 0.86 (0.84, 0.88) | NA | NA |
| | ≥ 1 symptom identified | 0.95 (0.91, 0.97) | 0.92 (0.91, 0.93) | 0.56 (0.45, 0.67) | 0.89 (0.87, 0.92) | NA | NA |

[1]. Data was not available.

too late in the infectious period due to mild symptoms, or no symptoms at all. Serology results may inform diagnoses of post-acute SARS-CoV-2 (PASC) and the appropriate treatment course, which may depend on whether patients are at increased risk for severe illness due to insufficient antibody response [35]. The reported sensitivity of the serology tests included in this analysis that were submitted for EUA approval were all >95% [36]. Our analysis of multiple large datasets of patients with confirmed SARS-CoV-2 infection suggests that serology tests performed lower than = expected–with PPA ranges (a measure analogous to sensitivity) from 65–90%.—Our results align with results from smaller, detailed laboratory evaluations that suggest a lack of harmonization, including optimization of cut-off values, may contribute to decreased overall performance. Additionally, our results align with studies that include more representative samples of milder or asymptomatic persons [37–39]. Two of seven tests

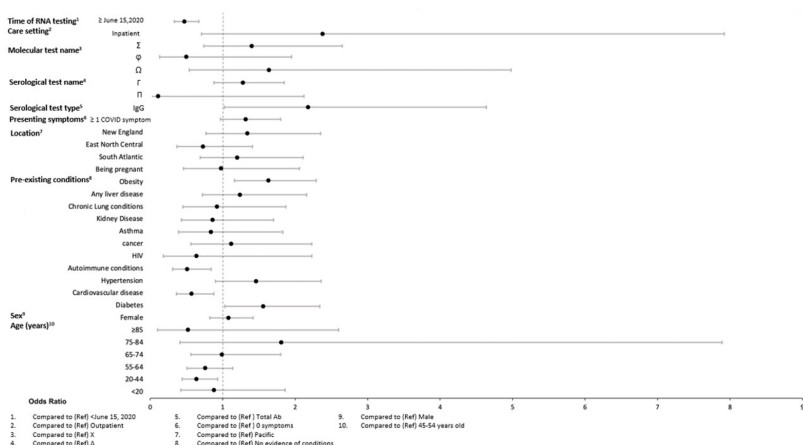

**Fig 4. Odds of seropositivity dataset A.**

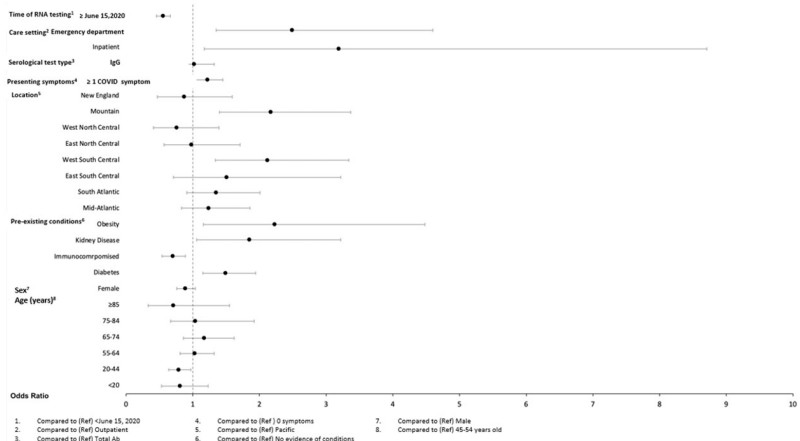

**Fig 5. Odds of seropositivity dataset B.**

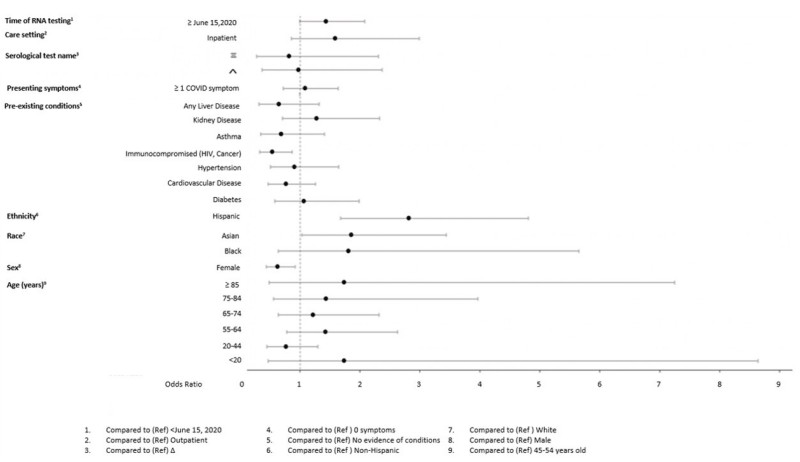

**Fig 6. Odds of seropositivity dataset C.**

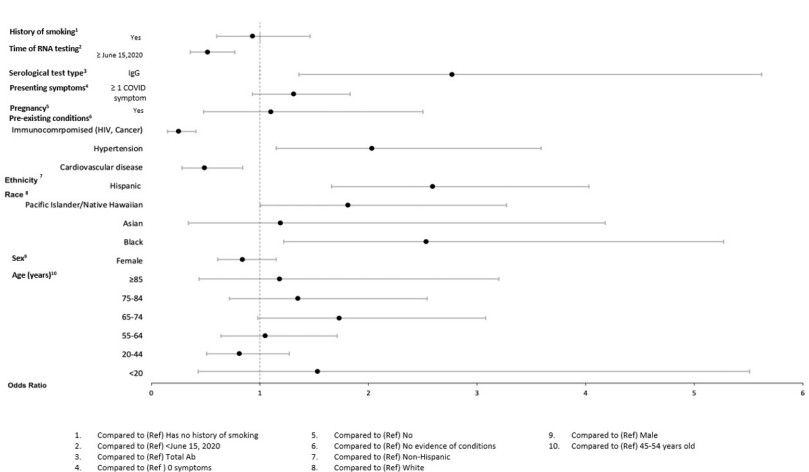

**Fig 7. Odds of seropositivity dataset D.**

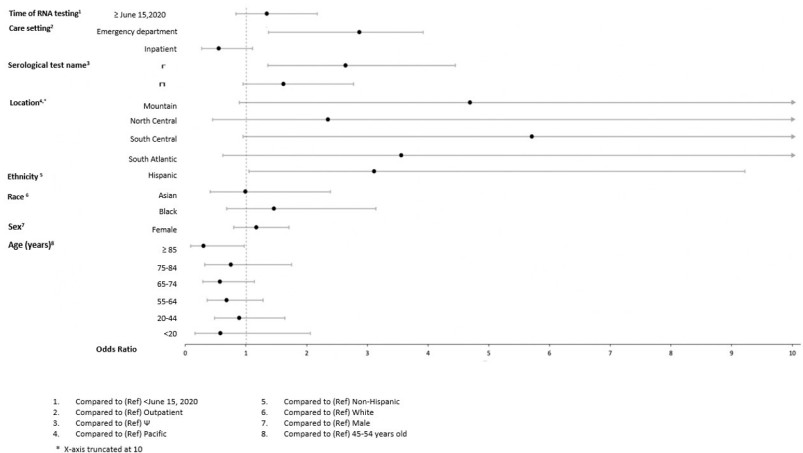

**Fig 8. Odds of seropositivity dataset E.**

reported across datasets achieved the EUA requirement of PPA ≥ 87%. As we did not have data on specific serology-molecular pairs or meta-information on the tests (including fidelity to protocols for serology and molecular test analysis), these results reflect more on the real-world implementation of the tests rather than the true quality of the tests. Specifically, where the same test was used across multiple datasets, they all performed similarly. For example, the serology test Γ performed similarly high (PPA >90%) across three datasets. However, the overall PPA for tests performed in datasets A and B were higher than in dataset E. A major factor that may have contributed to this difference is that the other serological tests reported to datasets A and B performed above the EUA requirement. In contrast, the other tests reported in dataset E performed below the EUA requirement. Additionally, datasets A and B leveraged administrative claims data and associated RNA and serology results with sample collection or sample receipt date, while dataset E associated results with the date the test was run.

Dataset E also represents those from a healthcare delivery system where serology tests were initially only used for symptomatic patients with at least 12 days of symptoms. This practice shifted after approximately two months (June 1, 2020) to a protocol that required both

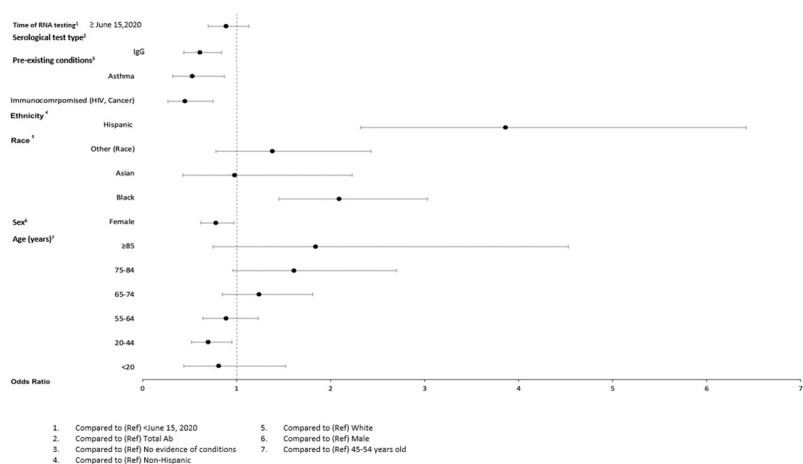

**Fig 9. Odds of seropositivity dataset F.**

molecular and serological testing for SARS-CoV-2 as part of pre-procedure screening. This protocol was in effect for another three months (August 31, 2020), after which the healthcare system shifted to unrestricted testing for both molecular and serology tests and saw a substantial drop in the use of serological testing. We expected that procedural "lags" to serotesting, combined with additional lags due to associating results with a date downstream from the clinical interaction, may have further extended the time between infection/symptom onset and the actual time of serology sampling. The impact of this misclassification may be most important for serology samples at the upper bounds of 90 days; where samples were likely >90 days from the point of infection and humoral antibodies more likely to have declined. Despite changes in the protocol over time, we observed no overall or test-specific difference in PPA before or since June 15, 2020 in dataset E. Nevertheless, administrative protocols create lags in serotesting that challenge our assumptions of whether the observed molecular "test date" is a good proxy for symptom onset. Absent any knowledge of such policy, it's difficult to make broad assumptions regarding patterns in molecular or serology testing unless established clinical protocols are known.

We observed that patients of Hispanic ethnicity compared to non-Hispanic patients, with pre-existing obesity and those who presented in the ED had a higher OR for seropositivity; and similarly higher PPA. These results further support what others have observed that persons with unmanaged diabetes, who are disproportionately people of color, are vulnerable to hyper-inflammation related to COVID-19 [40]. Furthermore, hyper-inflammation, including pro-inflammatory cytokine storm, has been associated with severe disease, reduced viral clearance [41], and sustained antibody production [42]. Although a recent small study showed that while a low viral load is associated with lower antibody response, clinical illness does not guarantee seroconversion [43]. Other studies have demonstrated people with cancer have a lower probability of mounting an immune response from the vaccine, as demonstrated by seroconversion, viral neutralization, and T-cell response [44, 45]. Our results demonstrating lower odds of seropositivity among those with cancer and other immunodeficiencies suggest that the same may be true regarding their antibody response to infection.

## Strengths

Our study has many strengths. This was a large assessment of serotesting across the U.S. in diverse datasets leveraging either EHR or claims data. We developed a protocol that incorporated the unique characteristics of each data source and provided a forum to transparently communicate and collaborate on study design and interpretation. We also established a platform to rapidly collect and analyze data from various systems to evaluate process improvement and identify important trends over time. Such a platform may be used to evaluate process improvement and comparisons within data systems. We did extensive characterization of missing data to guide model development and help with interpretation. Additionally, this study was conducted before public availability of COVID-19 vaccines across the U.S., which minimizes the potential for confounding related to vaccine-induced antibodies.

## Limitations

A major limitation in this real-world analysis is a large number of missing test names and relevant meta-data, including quality control measures adopted, for both molecular and serological tests. As such, we were unable to account for molecular-serology pairs when assessing PPA or the fidelity with which these tests were performed. A large amount of missing test name information limited our ability to describe trends by the manufacturer. Although, a thorough examination of missing data does not suggest differential missingness by age or sex.

Importantly, the intent of this analysis was not to evaluate individual tests, but the performance of serology in the context of real-world implementation of test protocols and varying reference standards. As discussed in our prior manuscript, the sample included in this study included those who were more likely to be serotested for SARS-CoV-2: White, 45–64 years of age, with prior history of cardiovascular disease. Nevertheless, there was still sufficiently large number of people to assess PPA trends among younger ages and in those with and without other pre-existing conditions. Finally, this study was conducted before the surge of the Omicron variant, which has been shown to have a number of mutations on the N-gene and S-gene that reduce the sensitivity of some diagnostic tests [46]. As such, our inference is limited to the SARS-CoV-2 variants prior to Omicron, primarily alpha.

## Conclusion

Across large samples of patients with molecularly confirmed SARS-CoV-2, serology tests did not consistently meet the EUA requirement of PPA $\geq$ 87% in the post-market setting. However, given the limited availability of test names, this analysis serves as a signal that further investigation into how serology and molecular tests are used, including protocol fidelity, is needed to understand ways to improve the real-world performance of serology tests.

Despite differences in testing protocols and data availability, the similarity in performance of serology tests across datasets suggests that serology tests were robust to differences in care settings. However, the real-world PPA for several serology tests did not meet EUA requirements; and the exclusive representation and low use of such tests in certain datasets look to have impacted the overall performance of serology tests in those datasets. Where data were sufficiently robust, we observed that people of Hispanic ethnicity had a higher odd of seropositivity than non-Hispanics. Higher odds of seropositivity in those with pre-existing diabetes or obesity further support the hypothesis that these conditions are associated with more severe disease, reduced viral clearance, and the sustained presence of antibodies. Conversely, lower odds of seropositivity among those with cancer and other immunodeficiencies suggest that immunopathology in these groups associated with the vaccine may extend to infection.

Interpreting results from real-world data collected from clinical and administrative databases is challenging. A clear understanding of testing protocols at the point of care is needed to validate assumptions regarding proxy variables and to interpret results. Incomplete information on race/ethnicity and test name limited our ability to address racial disparities in testing and real-world performance of serological tests. Nevertheless, implementing best practices for analyzing and reporting results from observational data across multiple datasets yields confidence in trends that are repeated. And where results are divergent, we were able to explore how differences in data sources may explain findings and target areas for future investigation. Improved data interoperability to link test names and clinical/demographic data is critical to enable rapid assessment of the real-world performance of in vitro diagnostic tests, particularly in the face of fast-mutating pathogens.

## Supporting information

**S1 Fig. Study design diagram dataset A.**
(TIF)

**S2 Fig. Study design diagram dataset B.**
(TIF)

**S3 Fig. Study design diagram dataset C.**
(TIF)

**S4 Fig. Study design diagram dataset D.**
(TIF)

**S5 Fig. Study design diagram dataset E.**
(TIF)

**S6 Fig. Study design diagram dataset F.**
(TIF)

**S1 Table. Characteristics of participating data sources and representative populations.**
(DOCX)

**S2 Table. Phenotype (code-lists) for specified presenting symptoms & pre-existing conditions.**
(DOCX)

## Acknowledgments

Special thanks to our advisors on this project from the U.S. Food and Drug Administration: Aloka Chakravarty, Tamar Lasky, Gina Valo, Mary Jung, Stephen Lovell, Jacqueline M Major, Daniel Caños, Sara Brenner, and Wendy Rubinstein; and Duke-Margolis: Christina Silcox. We thank all members of the Evidence Accelerator Workgroup for their support and feedback: Roland Romero, James Okusa, Elijah Mari Quinicot, Amar Bhat, Susan Winckler, Alecia Clary, Sadiqa Mahmood, Philip Ballentine, Perry L. Mar, Cynthia Lim Louis, Connor McAndrews, Elitza S. Theel, Cora Han, Pagan Morris, and Charles Wilson. A special thanks and recognition for the contributions and sacrifice of Dr. Michael Waters, our dear colleague, and friend who will be forever in our thoughts. We thank Amir Alishahi Tabriz MD, PhD for his assistance with manuscript preparation.

## Author Contributions

**Conceptualization:** Carla V. Rodriguez-Watson, Kanwal Ghauri.

**Data curation:** Carla V. Rodriguez-Watson, Carly Kabelac, Christopher M. Frederick, Natalie E. Sheils, Elizabeth H. Eldridge, Nancy D. Lin, Benjamin D. Pollock, Jennifer L. Gatz, Shaun J. Grannis, Rohit Vashisht, Kanwal Ghauri, Camille Knepper, Sandy Leonard, Peter J. Embi, Garrett Jenkinson, Reyna Klesh, Omai B. Garner, Ayan Patel, Lisa Dahm, Aiden Barin, Dan M. Cooper, Tom Andriola, Carrie L. Byington, Bridgit O. Crews, Atul J. Butte, Jeff Allen.

**Formal analysis:** Carla V. Rodriguez-Watson.

**Funding acquisition:** Carla V. Rodriguez-Watson.

**Investigation:** Carla V. Rodriguez-Watson, Carly Kabelac, Christopher M. Frederick, Natalie E. Sheils, Elizabeth H. Eldridge, Nancy D. Lin, Benjamin D. Pollock, Jennifer L. Gatz, Shaun J. Grannis, Rohit Vashisht, Kanwal Ghauri, Camille Knepper, Sandy Leonard, Garrett Jenkinson, Reyna Klesh, Omai B. Garner, Ayan Patel, Lisa Dahm, Aiden Barin, Dan M. Cooper, Tom Andriola, Carrie L. Byington, Bridgit O. Crews, Atul J. Butte, Jeff Allen.

**Methodology:** Carla V. Rodriguez-Watson, Anthony M. Louder, Christopher M. Frederick, Natalie E. Sheils, Elizabeth H. Eldridge, Nancy D. Lin, Benjamin D. Pollock, Jennifer L. Gatz, Shaun J. Grannis, Rohit Vashisht, Kanwal Ghauri, Camille Knepper, Sandy Leonard, Peter J. Embi, Garrett Jenkinson, Reyna Klesh, Omai B. Garner, Ayan Patel, Lisa Dahm,

Aiden Barin, Dan M. Cooper, Tom Andriola, Carrie L. Byington, Bridgit O. Crews, Atul J. Butte, Jeff Allen.

**Project administration:** Carla V. Rodriguez-Watson, Kanwal Ghauri.

**Resources:** Carla V. Rodriguez-Watson, Carly Kabelac, Kanwal Ghauri, Peter J. Embi.

**Software:** Carla V. Rodriguez-Watson, Kanwal Ghauri.

**Supervision:** Carla V. Rodriguez-Watson.

**Validation:** Carla V. Rodriguez-Watson.

**Visualization:** Carla V. Rodriguez-Watson.

**Writing – original draft:** Carla V. Rodriguez-Watson, Kanwal Ghauri.

**Writing – review & editing:** Carla V. Rodriguez-Watson, Anthony M. Louder, Carly Kabelac, Christopher M. Frederick, Natalie E. Sheils, Elizabeth H. Eldridge, Nancy D. Lin, Benjamin D. Pollock, Jennifer L. Gatz, Shaun J. Grannis, Rohit Vashisht, Kanwal Ghauri, Camille Knepper, Sandy Leonard, Peter J. Embi, Garrett Jenkinson, Reyna Klesh, Omai B. Garner, Ayan Patel, Lisa Dahm, Aiden Barin, Dan M. Cooper, Tom Andriola, Carrie L. Byington, Bridgit O. Crews, Atul J. Butte, Jeff Allen.

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
