## [Decision Letter · Decision Letter 0]

15 Jul 2022

PONE-D-22-11773Real-World Performance of SARS-Cov-2 Serology Tests in The United States, 2020.PLOS ONE

Dear Dr. Rodriguez-Watson,

Thank you for submitting your manuscript to PLOS ONE. After careful consideration, we feel that it has merit but does not fully meet PLOS ONE’s publication criteria as it currently stands. Therefore, we invite you to submit a revised version of the manuscript that addresses the points raised during the review process.

The study offers important information on the reliability of the serodiagnosis. Please see the comments from the reviewers and I hope you will find it helpful to improve the quality of the manuscript overall. it is indeed difficult to comprehend some of the images in the pdf version of the article. Please submit figures as recommended by plosOne (authors instructions) and plosOne can generate a pdf including the figures which are generally high quality. Based on the reveiwer's comments I am recommending your article for major revision and will look forward for the revised manuscript.

We look forward to receiving your revised manuscript.

Kind regards,

Padmapriya P Banada, PhD

Academic Editor

PLOS ONE

Journal Requirements:

“AJB is a co-founder and consultant to Personalis and NuMedii; consultant to Samsung, Mango Tree Corporation, and in the recent past, 10x Genomics, Helix, Pathway Genomics, and Verinata (Illumina); has served on paid advisory panels or boards for Geisinger Health, Regenstrief Institute, Gerson Lehman Group, AlphaSights, Covance, Novartis, Genentech, Merck, and Roche; is a shareholder in Personalis and NuMedii; is a minor shareholder in Apple, Facebook, Alphabet (Google), Microsoft, Amazon, Snap, Snowflake, 10x Genomics, Illumina, Nuna Health, Assay Depot (Scientist.com), Vet24seven, Regeneron, Sanofi, Royalty Pharma, Pfizer, BioNTech, AstraZeneca, Moderna, Biogen, Twist Bioscience, Pacific Biosciences, Editas Medicine, Invitae, Doximity, and Sutro, and several other non-health related companies and mutual funds; and has received honoraria and travel reimbursement for invited talks from Johnson and Johnson, Roche, Genentech, Pfizer, Merck, Lilly, Takeda, Varian, Mars, Siemens, Optum, Abbott, Celgene, AstraZeneca, AbbVie, Westat, several investment and venture capital firms, and many academic institutions, medical or disease specific foundations and associations, and health systems. AJB receives royalty payments through Stanford University, for several patents and other disclosures licensed to NuMedii and Personalis. AJB’s research has been funded by NIH, Northrup Grumman (as the prime on an NIH contract), Genentech, Johnson and Johnson, FDA, Robert Wood Johnson Foundation, Leon Lowenstein Foundation, Intervalien Foundation, Priscilla Chan and Mark Zuckerberg, the Barbara and Gerson Bakar Foundation, and in the recent past, the March of Dimes, Juvenile Diabetes Research Foundation, California Governor’s Office of Planning and Research, California Institute for Regenerative Medicine, L’Oreal, and Progenity.       

CLB has intellectual property in and receives royalties from BioFire, Inc. She serves as a scientific advisor to IDbyDNA (San Francisco, CA and Salt Lake City, UT); and is on the Board of the Commonwealth Fund.    

CK is a paid employee of Aetion and hold Aetion stock options.

NES is an employee of Optum Labs and owns stock in the parent company UnitedHealth group.

NDL was an employee of Health Catalyst at the time the work was performed.

JLG is a full-time employee of Regenstrief Institute, which provides independent research services to entities including those within the pharmaceutical and medical device industries.

SJG serves as Chief Medical Information Officer for the Indiana Health Information Exchange, and is a founding partner of Uppstroms, LLC.”

“Financial support for this work was provided in part by a grant from The Rockefeller Foundation.”

“CVRW was funded by a grant from the Rockefeller Foundation.  

BDP, CK, GJ used funding provided by Yale University-Mayo Clinic Center of Excellence in Regulatory Science and Innovation (CERSI), a joint effort between Yale University, Mayo Clinic, and the U.S. Food and Drug Administration (FDA) (3U01FD005938).           

CK, CMF, SJG, PJE, EHE, NDL, and JLG work was funded by a designated sub-grant from the FDA Foundation.

AJB funded by award number A128219 and Grant Number U01FD005978 from the FDA, which supports the UCSF-Stanford Center of Excellence in Regulatory Sciences and Innovation. Its contents are solely the responsibility of the authors and do not necessarily represent the official views of the HHS or FDA.

7. One of the noted authors is a group or consortium ”Evidence Accelerator Workgroup”. In addition to naming the author group, please list the individual authors and affiliations within this group in the acknowledgments section of your manuscript. Please also indicate clearly a lead author for this group along with a contact email address.

8. We note that Figure 1  in your submission contain map images which may be copyrighted. All PLOS content is published under the Creative Commons Attribution License (CC BY 4.0), which means that the manuscript, images, and Supporting Information files will be freely available online, and any third party is permitted to access, download, copy, distribute, and use these materials in any way, even commercially, with proper attribution. For these reasons, we cannot publish previously copyrighted maps or satellite images created using proprietary data, such as Google software (Google Maps, Street View, and Earth). For more information, see our copyright guidelines: http://journals.plos.org/plosone/s/licenses-and-copyright.

Reviewers' comments:

Reviewer's Responses to Questions

**Comments to the Author**

1. Is the manuscript technically sound, and do the data support the conclusions?

Reviewer #1: Yes

Reviewer #2: Yes

2. Has the statistical analysis been performed appropriately and rigorously? 

Reviewer #1: No

Reviewer #2: Yes

3. Have the authors made all data underlying the findings in their manuscript fully available?

Reviewer #1: Yes

Reviewer #2: Yes

4. Is the manuscript presented in an intelligible fashion and written in standard English?

Reviewer #1: Yes

Reviewer #2: Yes

5. Review Comments to the Author

Reviewer #1: The Rodriguez-Watson et al. manuscript describes a synthesis of real-world serology testing for SARS-CoV-2. It analyses agreements between SARS-CoV-2 PCR testing and antibody detection assays in 6 US health systems across different setting (inpatient, outpatient, ED, long-term care). The aims of the manuscript were to address gaps in understanding exposure to SARS-CoV-2 and identifying factors associated with seroconversion.

The manuscript is well written and understandable even though presents a large amount of data.

Although the odd of seropositivity according to demographics gives a good understanding of factors that might affect seroconversion, the authors failed to address the gaps in understanding exposure.

Comments:

1- The manuscript describes agreement (PPA) between PCR testing and serology results at 14-90 days post PCR. The approach to analyzing agreement is not complete with no mention of kappa (Cohen, McNemar test, etc.).

2- Is there a loss of agreement depending on day serology was done?

3- When comparing PPA observed according to ethnicity or other factors, are the differences significant?

4- Line 284, the authors mention test quality. Many studies looking at the quality of several serology tests used in this manuscript have been published, can the authors discuss further and refer to these manuscripts? Does lower agreement observe in this study match the published data?

4- In the section results, no odd ratio is given, only 95%CI. Please add.

5- Factors associated with seropositivity, line 257-266. Are the differences described significant?

6- Line 135 change e.g. by i.e. or remove altogether

7-Line 135, remove comma after IgG, “IgG, [21],”

8- Table 1, “Na” in lowercase, all other table “NA” is capitalised. Please homogenise throughout.

9- Line 313, should read sustained antibody production.

10- Line 359, should read the sustained presence of antibodies.

11- Line 371, should be “in vitro”.

Reviewer #2: Summary:

In this important study, Dr. Rodriguez-Watson and colleagues studied 6 large-scale datasets to understand the 2020 performance of real-world use of EUA-approved SARS-CoV-2 serology testing after a positive molecular test. The group demonstrates substantial real-world variance in the PPA of these tests across health contexts.

Major comments

1. What was done statistically regarding individuals with a 2nd serology test? Line 161 implies only the first test was used. What was the concordance/timing between 1st and 2nd tests? Did 2nd tests, where done, have a different PPA?

2. Dataset C seems to have a broadly lower PPA vs the other datasets, has the smallest N, and is relatively geographically restricted. This dataset does not appear to have any manufacturer molecular test names available, but there is no PPA reported in the Unknown/missing category in table 2 for that variable. Is this dataset usable? It would seem that as both the serological and molecular test characteristics would contribute to the PPA, not knowing the molecular test name at all makes using this dataset problematic.

3. What is known about the contribution of “other” molecular tests to this dataset, such as the adoption of “rapid” PCR testing and “in-house” testing that some institutions produced during this time period? Is it possible to address those tests where both the serological and molecular tests are known? As above, the confounding factor of molecular test characteristics could influence the PPA of the serological tests in question.

4. Time between the molecular and serological test seems like a key point as well. Do you have that data? Does time between tests affect the results?

Minor comments

1. The figures are not showing well (scattered pixels) in my Adobe Acrobat Pro DC view of the PDF. Please ensure that high quality images are used during publication. I can access the tif files, which look right.

2. I do not understand why figures 2-6 list the study period as starting in 2019. Is this a typo or is this correct? The methods list 2020 as start date, which would make sense given the dates of the pandemic.

3. Please check line 380, there may be a comma instead of a period.

4. Perhaps figures 2-7 should be combined into a single summary figure (with total N for the entire study) and the individual flowsheets by cohort might be moved to supplemental. The reader might better grasp the overall study with a simpler summary figure.

6. PLOS authors have the option to publish the peer review history of their article (what does this mean?). If published, this will include your full peer review and any attached files.

Reviewer #1: No

Reviewer #2: No

---

## [Author Response · Author response to Decision Letter 0]

3 Nov 2022

Dear Dr. Padmapriya Banada:

We thank you and the reviewers for your thoughtful comments on our manuscript. We appreciate the opportunity to respond and believe the revisions have improved the manuscript. Below, please find a table with the summary of reviewer comments and our responses. Please be advised that all line numbers referenced in the responses below correspond to line numbers in the tracked changes version of the manuscript. Please reach out with any additional questions, or if more clarification is required. We look forward to hearing from you.

Regards, 

Carla Rodriguez-Watson, PhD, MPH

Director of Research

Reagan-Udall Foundation for the FDA

Comments/Questions Response

Journal Requirements

 Please see the new version with the correct requirements. Additionally, we have listed manuscript requirements below with the acknowledgment that they are complete. 

Journal Requirement Acknowledgement of Completion 

Article Title YES

Author Byline YES

Affiliations YES

Corresponding Authorship YES

Contributorship YES

Consortia or other Group Authors YES

Level 1 Heading YES

Figure Citations YES

Figure Captions YES

File Naming for Figures YES

Display/Numbered Equation N/A

Inline Equation N/A

Level 2 Heading YES

Level 3 heading YES

Please submit your manuscript in double-space paragraph format. YES

Tables and Table Citations YES

Reference Citations YES

Supporting Information Citations YES

Acknowledgments- No funding or competing interest information YES

References YES

Supporting Information Captions YES

File Naming for Supporting Information YES

We have listed our full ethics statement in lines 132-136 of the methods section: “The Western - Copernicus Group (WCG) Institutional Review Board (IRB), the IRB of record for the Reagan-Udall Foundation for the FDA, reviewed the study and determined it to be non-human subjects research. Additionally, all legal and ethical approvals for use of the data included in this study were submitted, reviewed, and/or obtained locally at each contributing dataset by an IRB and/or governing board.”

Thank you for pointing this out. We have updated the sections with the correct grant numbers in the online submission. We have also included the correct grant numbers below.

Funding Information Section:

Financial support for this work was provided in part by a grant from The Rockefeller Foundation (HTH 030 GA-S).

BDP, CK, GJ used funding provided by Yale University-Mayo Clinic Center of Excellence in Regulatory Science and Innovation (CERSI), a joint effort between Yale University, Mayo Clinic, and the U.S. Food and Drug Administration (FDA) (3U01FD005938) (https://www.fda.gov/). 

CK, CMF, SJG, PJE, EHE, NDL, and JLG work was funded by a designated sub-grant from the FDA Foundation.

AJB funded by award number A128219 and Grant Number U01FD005978 from the FDA, which supports the UCSF-Stanford Center of Excellence in Regulatory Sciences and Innovation. Its contents are solely the responsibility of the authors and do not necessarily represent the official views of the HHS or FDA.

“CRW receives research funding from Novartis, Merck and AbbVie; and holds minor stock in Gilead. AJB is a co-founder and consultant to Personalis and NuMedii; consultant to Samsung, Mango Tree Corporation, and in the recent past, 10x Genomics, Helix, Pathway Genomics, and Verinata (Illumina); has served on paid advisory panels or boards for Geisinger Health, Regenstrief Institute, Gerson Lehman Group, AlphaSights, Covance, Novartis, Genentech, Merck, and Roche; is a shareholder in Personalis and NuMedii; is a minor shareholder in Apple, Facebook, Alphabet (Google), Microsoft, Amazon, Snap, Snowflake, 10x Genomics, Illumina, Nuna Health, Assay Depot (Scientist.com), Vet24seven, Regeneron, Sanofi, Royalty Pharma, Pfizer, BioNTech, AstraZeneca, Moderna, Biogen, Twist Bioscience, Pacific Biosciences, Editas Medicine, Invitae, Doximity, and Sutro, and several other non-health related companies and mutual funds; and has received honoraria and travel reimbursement for invited talks from Johnson and Johnson, Roche, Genentech, Pfizer, Merck, Lilly, Takeda, Varian, Mars, Siemens, Optum, Abbott, Celgene, AstraZeneca, AbbVie, Westat, several investment and venture capital firms, and many academic institutions, medical or disease specific foundations and associations, and health systems. AJB receives royalty payments through Stanford University, for several patents and other disclosures licensed to NuMedii and Personalis. AJB’s research has been funded by NIH, Northrup Grumman (as the prime on an NIH contract), Genentech, Johnson and Johnson, FDA, Robert Wood Johnson Foundation, Leon Lowenstein Foundation, Intervalien Foundation, Priscilla Chan and Mark Zuckerberg, the Barbara and Gerson Bakar Foundation, and in the recent past, the March of Dimes, Juvenile Diabetes Research Foundation, California Governor’s Office of Planning and Research, California Institute for Regenerative Medicine, L’Oreal, and Progenity. 

CLB has intellectual property in and receives royalties from BioFire, Inc. She serves as a scientific advisor to IDbyDNA (San Francisco, CA and Salt Lake City, UT); and is on the Board of the Commonwealth Fund. 

CK is a paid employee of Aetion and hold Aetion stock options.

NES is an employee of Optum Labs and owns stock in the parent company UnitedHealth group.

NDL was an employee of Health Catalyst at the time the work was performed.

JLG is a full-time employee of Regenstrief Institute, which provides independent research services to entities including those within the pharmaceutical and medical device industries.

SJG serves as Chief Medical Information Officer for the Indiana Health Information Exchange, and is a founding partner of Uppstroms, LLC.”

Thank you for pointing this out. There are no conflicts between competing interests and PLOS One policy on data sharing. We have revised our competing interest in the cover letter to include the requested statement below:

This does not alter our adherence to PLOS ONE policies on sharing data and materials.

“Financial support for this work was provided in part by a grant from The Rockefeller Foundation.”

We note that you have provided additional information within the Acknowledgements Section that is not currently declared in your Funding Statement. Please note that funding information should not appear in the Acknowledgments section or other areas of your manuscript. We will only publish funding information present in the Funding Statement section of the online submission form. Please remove any funding-related text from the manuscript and let us know how you would like to update your Funding Statement. 

Currently, your Funding Statement reads as follows:

“CVRW was funded by a grant from the Rockefeller Foundation. 

BDP, CK, GJ used funding provided by Yale University-Mayo Clinic Center of Excellence in Regulatory Science and Innovation (CERSI), a joint effort between Yale University, Mayo Clinic, and the U.S. Food and Drug Administration (FDA) (3U01FD005938). 

CK, CMF, SJG, PJE, EHE, NDL, and JLG work was funded by a designated sub-grant from the FDA Foundation.

AJB funded by award number A128219 and Grant Number U01FD005978 from the FDA, which supports the UCSF-Stanford Center of Excellence in Regulatory Sciences and Innovation. Its contents are solely the responsibility of the authors and do not necessarily represent the official views of the HHS or FDA.

Thank you for bringing this to our attention. We have removed the funding information from the Acknowledgement section and include all funding information as updated above under “Funding Information.”

Thank you for bring this to our attention. The data that we used for our findings are not bound to some legal or ethical restrictions. All relevant data are contained within the manuscript or its supporting documents. Person-level data are unavailable. Qualified researchers interested in accessing deidentified person-level data should contact the corresponding author for more

Information. 

7. One of the noted authors is a group or consortium “Evidence Accelerator Workgroup”. In addition to naming the author group, please list the individual authors and affiliations within this group in the acknowledgments section of your manuscript. Please also indicate clearly a lead author for this group along with a contact email address.

Thank you for bringing this to our attention. The Evidence Accelerator Workgroup refers to our consortium. We only had included those who met the ICMJE authorship as co-authors, but wanted to acknowledge all those who worked behind the scenes to support the work. Given this advice, we will include their names as acknowledgements. We have added the following phrase “We thank all members of the Evidence Accelerator Workgroup for their support and feedback: [list names]” to lines 395-399.

8. We note that Figure 1 in your submission contain map images which may be copyrighted. All PLOS content is published under the Creative Commons Attribution License (CC BY 4.0), which means that the manuscript, images, and Supporting Information files will be freely available online, and any third party is permitted to access, download, copy, distribute, and use these materials in any way, even commercially, with proper attribution. For these reasons, we cannot publish previously copyrighted maps or satellite images created using proprietary data, such as Google software (Google Maps, Street View, and Earth). For more information, see our copyright guidelines: http://journals.plos.org/plosone/s/licenses-and-copyright.

Thank you for bringing this to our attention. We had previously attached the Content Permission Form as part of the submission package. We attached the Content Permission Form as part of the submission package again and added the reprint information to the figure 1’s caption.

Reviewer 1 comments:

1. The manuscript describes agreement (PPA) between PCR testing and serology results at 14-90 days post PCR. The approach to analyzing agreement is not complete with no mention of kappa (Cohen, McNemar test, etc.). 

Thank you for the comment. This analysis focused on serology agreement with positive RNA tests. As such, we did not collect any RNA negative results, which are required to assess kappa. 

2. Is there a loss of agreement depending on day serology was done? 

Yes, there was loss of agreement depending on the day serology test was done, and one of the results we intend to present in a subsequent publication. The current manuscript is intended to discuss the overall agreement between serology and PCR during the window for which agreement is most expected (days 14-90 after PCR) and by key demographic and clinical factors. A subsequent publication will look at how that agreement changes over different time periods. 

3. When comparing PPA observed according to ethnicity or other factors, are the differences significant? 

We report differences in PPA by race and ethnicity only where there are sufficient data (i.e. missing <30% and sample size ≥ n=40). We consistently observed higher PPA in Hispanic ethnicity compared to non-Hispanic ethnicity, as demonstrated by complete separation of confidence intervals (Clopper-Pearson). We did not conduct direct comparison of PPA across groups. We clarified the statistical analysis section (lines 171-173) to describe the meaning of 'significant differences’ outside of direct comparisons: “We calculated exact (Clopper-Pearson) 95% confidence intervals (CI). We report significant differences where 95% CI have complete separation - although we did not conduct formal statistical comparisons of PPA between groups.”

4. Line 284, the authors mention test quality. Many studies looking at the quality of several serology tests used in this manuscript have been published, can the authors discuss further and refer to these manuscripts? Does lower agreement observe in this study match the published data?

Thank you for the great suggestion. We have now included in the Discussion a comparison of our PPA results to those from sensitivities reported in original EUA submissions; as well as in context of other external evaluations (see lines 290-298):

“The reported sensitivity of the serology tests included in this analysis that were submitted for EUA approval were all >95% [36]. Our analysis of multiple large datasets of patients with confirmed SARS-CoV-2 infection suggests that serology tests performed lower than =expected – with PPA ranges (a measure analogous to sensitivity) from 65-90%. - Our results align with results from smaller, detailed laboratory evaluations that suggest a lack of harmonization, including optimization of cut-off values, may contribute to decreased overall performance. Additionally, our results align with studies that include more representative samples of milder or asymptomatic persons [37–39].” Although many of these evaluations are still limited by much smaller sample sizes than we report (though they are more detailed laboratory studies) and appear limited in the replicability of the same assay result across different laboratories, they do contain more diverse populations and note lower performance compared to initial certification related evaluations. 

5. In the section results, no odd ratio is given, only 95%CI. Please add. 

Apologies for the confusion, the results given in the parenthesis in this section are actually ranges of OR, not the 95% CIs. We have clarified this in reporting results in this version. 

6. Factors associated with seropositivity, line 257-266. Are the differences described significant?

Thank you for the comment. The differences described in the text were significant. Significant differences can be observed in the figures as those whose 95% CI does not cross the ‘1’on the X axis. Across the board, we found age 20-44 yrs to have a lower odds of seropositivity than those 45-54 yrs; Hispanic Ethnicity to have higher odds than non-Hispanic; immunocompromised to have a lower odds of seropositivity than those with no pre-existing conditions. ORs for obesity and presenting with >1 COVID symptom also were also significantly elevated in >1 data source. We have clarified this in the text by adding the word “significantly” to lines 269 and 273.

7. Line 135 change e.g. by i.e. or remove altogether

Thank you for calling this out. To clarify, PCR was not the only molecular conducted. The list included NAAT, RT-PCR so we respectfully leave it as “e.g”

8. Line 135, remove comma after IgG, “IgG, [21],” 

Thank you for bring this to our attention. We have removed the comma after IgG. Please see line 141. 

9. Table 1, “Na” in lowercase, all other table “NA” is capitalised. Please homogenise throughout.

 Thank you for bring this to our attention. 

We have updated Table 1 to read NA instead of Na to match the other table. 

10. Line 313, should read sustained antibody production.

Thank you for bring this to our attention. We have updated the language in line 331. 

11. Line 359, should read the sustained presence of antibodies.

Thank you for bring this to our attention. We have updated the language in line 377-378. 

12. Line 371, should be “in vitro”. 

Thank you for bring this to our attention. We have updated the language in line 389. 

Reviewer #2 comments:

1. What was done statistically regarding individuals with a 2nd serology test? Line 161 implies only the first test was used. What was the concordance/timing between 1st and 2nd tests? Did 2nd tests, where done, have a different PPA?

The majority of person in our cohort had just one serology test done. In order to compare consistently, we picked the first test done on an individual occurring 14 or more days after their positive molecular test. This choice has the added benefit of avoiding bias that would occur if we counted the same individual more than once knowing that individuals are more likely to retest if they get a result which is unexpected.

2. Dataset C seems to have a broadly lower PPA vs the other datasets, has the smallest N, and is relatively geographically restricted. This dataset does not appear to have any manufacturer molecular test names available, but there is no PPA reported in the Unknown/missing category in table 2 for that variable. Is this dataset usable? It would seem that as both the serological and molecular test characteristics would contribute to the PPA, not knowing the molecular test name at all makes using this dataset problematic. 

Keen observations! Not all partners reported the name of the molecular test and thus, did not estimate PPA by molecular test (reference). All partners did estimate PPA by serology test. We acknowledge in the limitations that we did not analyze molecular-serology pairs. As you note, characteristics of each test may influence PPA results, though we account for many other factors that may affect results. Because of this limitation, we chose not to report results by specific test name as it may suggest a deficiency that we could not accurately explain. 

3. What is known about the contribution of “other” molecular tests to this dataset, such as the adoption of “rapid” PCR testing and “in-house” testing that some institutions produced during this time period? Is it possible to address those tests where both the serological and molecular tests are known? As above, the confounding factor of molecular test characteristics could influence the PPA of the serological tests in question.

Rapid tests were not included in this analysis. One site included an ‘in-house’ test that was not FDA approved or for whom an EUA was not issued; the majority used only FDA approved or EUA tests. As such, we did not conduct the suggested analysis. 

4. Time between the molecular and serological test seems like a key point as well. Do you have that data? Does time between tests affect the results?

Yes, we agree with your comment. Analysis of PPA since the time of the molecular test is the focus of a subsequent manuscript. In the current analysis, we focus on tests 14-90 days from positive molecular test to maximize sensitivity of the test. 

5. The figures are not showing well (scattered pixels) in my Adobe Acrobat Pro DC view of the PDF. Please ensure that high quality images are used during publication. I can access the tif files, which look right.

Thank you for letting us know. We will resubmit the figures as .tiff files. 

6. I do not understand why figures 2-6 list the study period as starting in 2019. Is this a typo or is this correct? The methods list 2020 as start date, which would make sense given the dates of the pandemic.

Thank you for the question. This is correct. March 2019 represents the collection of baseline data such as comorbidities and socioeconomic data. The study follow-up period begins in March 1, 2020 and continues until September 30, 2020.

7. Please check line 380, there may be a comma instead of a period.

Thank you for bring this to our attention. We have removed the comma and changed it to a period. Please see line 399

8. Perhaps figures 2-7 should be combined into a single summary figure (with total N for the entire study) and the individual flowsheets by cohort might be moved to supplemental. The reader might better grasp the overall study with a simpler summary figure. 

Thank you for the suggestion. We created a summary figure (Fig 2) that depicts the general study design and the sample size of each partners study cohort; thus renumbering the remaining figures and moving the individual study diagrams to the supplemental figures. Each of the final cohorts (A-F) include patients that have both molecular test and follow up serology test as indicated in the methods. The parallel analysis approach entails that each cohort was analyzed separately according to a common protocol. Since this was not an aggregated analysis, we did not aggregate the numbers across partners.

---

## [Decision Letter · Decision Letter 1]

19 Dec 2022

Real-world performance of SARS-Cov-2 serology tests in the United States, 2020.

PONE-D-22-11773R1

Dear Dr. Rodriguez-Watson,

We’re pleased to inform you that your manuscript has been judged scientifically suitable for publication and will be formally accepted for publication once it meets all outstanding technical requirements.

Kind regards,

Padmapriya P Banada, PhD

Academic Editor

PLOS ONE

Additional Editor Comments (optional):

Thank you for resubmitting your article addressing the comments raised by the reviewers. Thank you for considering the comments constructive. The manuscript is greatly improved and is clear.

Reviewers' comments:

Reviewer's Responses to Questions

**Comments to the Author**

1. If the authors have adequately addressed your comments raised in a previous round of review and you feel that this manuscript is now acceptable for publication, you may indicate that here to bypass the “Comments to the Author” section, enter your conflict of interest statement in the “Confidential to Editor” section, and submit your "Accept" recommendation.

Reviewer #1: All comments have been addressed

Reviewer #2: All comments have been addressed

2. Is the manuscript technically sound, and do the data support the conclusions?

Reviewer #1: Yes

Reviewer #2: Yes

3. Has the statistical analysis been performed appropriately and rigorously? 

Reviewer #1: Yes

Reviewer #2: Yes

4. Have the authors made all data underlying the findings in their manuscript fully available?

Reviewer #1: Yes

Reviewer #2: Yes

5. Is the manuscript presented in an intelligible fashion and written in standard English?

Reviewer #1: Yes

Reviewer #2: Yes

6. Review Comments to the Author

Reviewer #1: Thank you for adressing all the comments.

Two very minor typos if this can be changed before publication

line 111 : Figs 2 - remove s

line 161 add space between CI and reference [33]

Reviewer #2: Thank you for addressing my comments and questions. I have no additional concerns or questions at this time. I recommend acceptance and publication.

7. PLOS authors have the option to publish the peer review history of their article (what does this mean?). If published, this will include your full peer review and any attached files.

Reviewer #1: No

Reviewer #2: No

---

## [Editor Report · Acceptance letter]

24 Jan 2023

PONE-D-22-11773R1 

Real-world performance of SARS-Cov-2 serology tests in the United States, 2020. 

Dear Dr. Rodriguez-Watson:

I'm pleased to inform you that your manuscript has been deemed suitable for publication in PLOS ONE. Congratulations! Your manuscript is now with our production department. 

Kind regards, 

on behalf of

Dr. Padmapriya P Banada 

Academic Editor

PLOS ONE